# Programmed spatial organization of biomacromolecules into discrete, coacervate-based protocells

Wiggert J. Altenburg [1,2], N. Amy Yewdall[1,2], Daan F. M. Vervoort[1,2], Marleen H. M. E. van Stevendaal[2,3], Alexander F. Mason[2,3 ✉] & Jan C. M. van Hest [1,2,3 ✉]

The cell cytosol is crowded with high concentrations of many different biomacromolecules, which is difficult to mimic in bottom-up synthetic cell research and limits the functionality of existing protocellular platforms. There is thus a clear need for a general, biocompatible, and accessible tool to more accurately emulate this environment. Herein, we describe the development of a discrete, membrane-bound coacervate-based protocellular platform that utilizes the well-known binding motif between $Ni^{2+}$-nitrilotriacetic acid and His-tagged proteins to exercise a high level of control over the loading of biologically relevant macromolecules. This platform can accrete proteins in a controlled, efficient, and benign manner, culminating in the enhancement of an encapsulated two-enzyme cascade and protease-mediated cargo secretion, highlighting the potency of this methodology. This versatile approach for programmed spatial organization of biologically relevant proteins expands the protocellular toolbox, and paves the way for the development of the next generation of complex yet well-regulated synthetic cells.

[1] Department of Biomedical Engineering, Eindhoven University of Technology, PO Box 513, 5600 MB Eindhoven, The Netherlands. [2] Institute for Complex Molecular Systems, Eindhoven University of Technology, PO Box 513, 5600 MB Eindhoven, The Netherlands. [3] Department of Chemical Engineering and Chemistry, Eindhoven University of Technology, PO Box 513, 5600 MB Eindhoven, The Netherlands. ✉email: a.f.Mason@tue.nl; j.c.m.v.hest@tue.nl

The bottom-up recreation of cellular processes into synthetic compartments has in recent years emerged as an exciting line of research to study biological processes in a controlled environment[1–4]. Historically, liposomes have been favored in the creation of photocellular compartments, encapsulating communication networks, and complex biochemical reactions, such as DNA-mediated self-replication, signaling cascades, and in vitro transcription and translation (IVTT)[5–7]. Other membrane-bound protocells, such as polymersomes[8], colloidosomes[9], and proteinosomes[10], are also able to encapsulate functional biomacromolecular systems within cell-mimetic capsules. However, these membrane-bound protocells share intrinsic weaknesses, such as low and heterogeneous encapsulation efficiencies, lack of control over stoichiometry of multimeric cargoes, and rudimentary release mechanisms, limiting their development into truly cell-mimetic platforms[11,12]. Microfluidic techniques can mitigate this to a degree[13,14] but are often difficult to implement compared to self-assembled systems, which rely on molecular interactions to form hierarchical structures.

Protocell models based on condensed aqueous droplets, such as complex coacervates, aqueous two-phase systems (ATPS), and liquid–liquid phase separation of intrinsically disordered proteins, could provide unique opportunities for the incorporation of additional functionalities. In these systems, attractive or segregative mechanisms drive the formation of polymer-rich, crowded cell-sized droplets, which typically exhibit strong incorporation of cargo into their core due to charge complementarity and/or hydrophobicity. Various examples have been reported in which these condensed droplets have been used as protocells capable of controlled growth[15], as tools to study synthetic organelles[16], as well as compartments in which the activity for ribozyme was enhanced[17] or IVTT could be performed[18,19]. In addition, the inherently crowded environment within membrane-free protocells resembles the interior of a living cell more closely than membrane-bound protocells, as ~30% of the cytoplasmic volume is occupied by biomacromolecules[20,21]. This is an important parameter to take into consideration, as crowding is reported to influence macromolecular association, protein conformation, and diffusional processes[22–26]. However, while cargo loading within membrane-free protocells is improved compared to membrane-bound systems, the loading mechanism is typically discriminate: there is only selective uptake of components in the coacervate phase when they are modified with a complementary charge or low complexity, intrinsically unstructured regions, such as LAF-1[27] or elastin[28], which limits the general applicability of these platforms[27–30]. For example, in our recently published work describing discrete terpolymer-stabilized complex coacervate protocells[31], the formation of stable coacervate protocells was demonstrated while protein uptake remained a challenge due to the coacervates' (positively) charged nature. As a result, highly negatively charged proteins, made by either amino acid mutations on the protein surface or via succinylation of surface accessible lysine residues into carboxylates, were readily taken up inside the coacervate phase, whereas neutral or positively charged proteins were excluded[32,33]. Succinylation is an effective way to obtain charge inversion, but can also easily result in a loss of function due to modification of functionally important lysine residues, or unfolding[34]. While surface modification via mutagenesis can stabilize proteins, it is a long and iterative process[35]. Moreover, only a limited number of proteins are reported that can handle multiple of these rigorous changes[36,37]. There is thus a clear need to develop a general, biologically compatible, and modular strategy to control the spatial organization of functional biomolecules into the cytosol-mimetic environment, and enable the expansion of life-like functions in bottom-up synthetic cells.

Herein we describe a robust method to recruit a broad range of recombinant proteins into coacervate-based protocells without extensive modification of the cargo, expanding the toolbox of possible biomimetic functionalities. To engineer this efficient uptake, additional amylose modified with nitrilotriacetic (NTA), which is well-known to complex $Ni^{2+}$ and reversibly bind polyhistidine-tagged (His-tagged) proteins, was synthesized and added to the coacervate (Fig. 1). The modified coacervate droplet was capable of efficient sequestration of different recombinant proteins, enabling control over their local concentration. This property was exploited to reconstitute a synthetic two-step enzymatic cascade, demonstrating the functional benefit of concentrating biologically active components by the enhancement in the activity. Lastly, we designed a protease-mediated release mechanism, enabling the controlled secretion of cargo from the coacervate core. Our strategy for the spatial organization of proteins into discrete self-assembled systems has enabled a broader scope of biologically relevant functionalities, as well as providing a guiding principle of non-covalent binding between the protocellular scaffold and the functional macromolecular cargo to other cell-mimetic platforms.

## Results

**Controlled protein uptake into coacervates.** In preceding work, coacervates were created via the coalescence of positively charged quaternized amylose (Q-Am) and negatively charged carboxymethylated amylose (Cm-Am) after which they were stabilized by the addition of a terpolymer (Fig. 1). Efficient uptake of macromolecular cargo inside the coacervate core during the formation process required a negative charge[32]. In order to overcome the limitation of charged cargo, the programmed uptake of recombinant proteins was devised by functionalizing amylose with an NTA group, which coordinates $Ni^{2+}$ and binds His-tagged proteins. NTA-amylose (NTA-Am) was synthesized from Cm-Am in a two-step reaction (Supplementary Fig. 2). The carboxylic acid moiety was first activated via EDC/NHS chemistry, followed by the addition of $N_\alpha,N_\alpha$-bis(carboxymethyl)-L-lysine hydrate under basic conditions to yield NTA-Am. In order to assess protein sequestration via this methodology, recombinant superfolder Green Fluorescent Protein with a cleavable histidine-tag (sfGFP-His) was chosen as a model protein, as this fluorescent protein is not sufficiently negatively charged to be taken up in the coacervate core via electrostatic interactions. sfGFP-His was mixed during protocell formation with the amylose polymers (2:0.8:0.2 mass ratio of Q:Cm:NTA) in a $Ni^{2+}$ containing buffer (PBS, pH:7.4, 7.5 μM $NiSO_4$). The uptake of protein was immediately noticeable inside the discrete coacervate protocells (Fig. 2a). As expected, the uptake efficiency was found to be dependent on the presence of the His-tag as well as the $Ni^{2+}$ ions, demonstrated by the absence of protein sequestration when either was omitted. In fact, adjusting the $Ni^{2+}$ concentration results in control over the protein uptake, with the maximum of protein loading observed at 7.5 μM $Ni^{2+}$, which corresponds well to the estimated 8.7 μM of NTA groups added (Supplementary Fig. 6). Importantly, sfGFP uptake into the coacervate core is homogeneous and independent of protocell size. This was clearly seen in confocal micrographs (Fig. 2a and Supplementary Fig. 4), and also confirmed by flow cytometry analysis (Fig. 2b), represented by a linear correlation between protocell volume and fluorescence intensity of sfGFP-His. At larger size there does appear to be a deviation in the linear behavior, this likely due to the fact that these larger protocells measured exceed the recommended instrumental limits[38]. The local sfGFP-His concentration inside the coacervate was assessed with the aid of an sfGFP-His fluorescence calibration curve. An initial bulk concentration of 250 nM

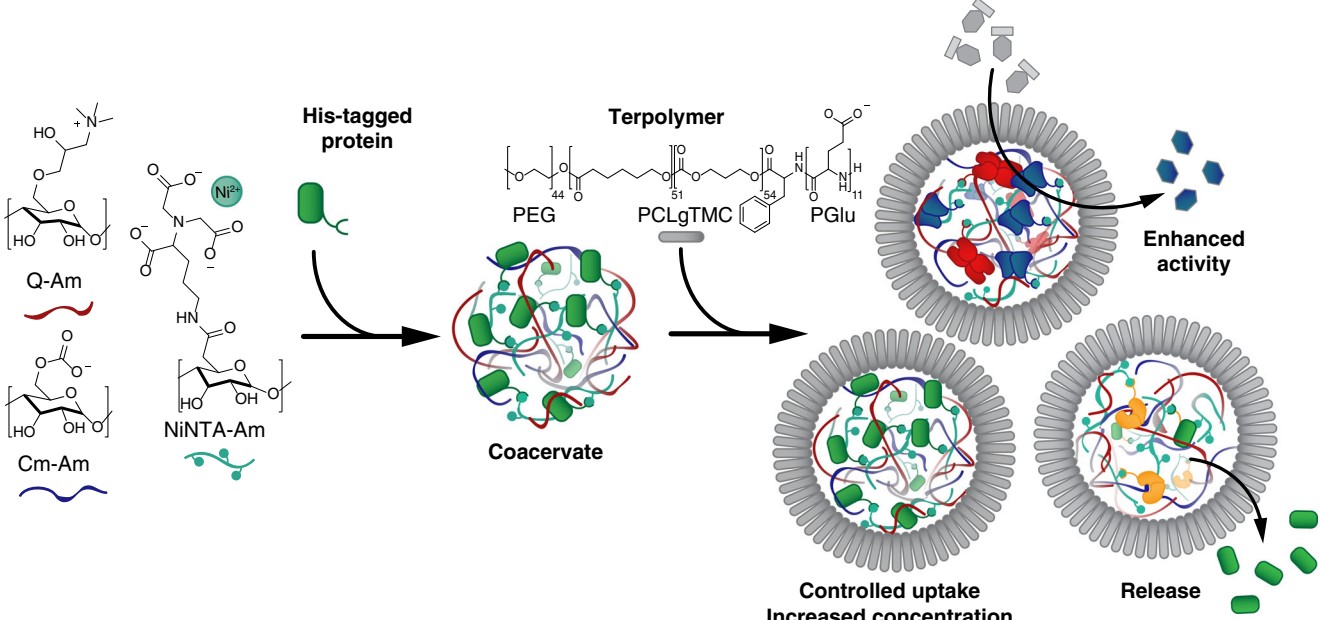

**Fig. 1 Schematic overview of chemically programmed loading of His-tagged proteins into Ni2+-NTA-functionalized protocells.** Q-Am (red), Cm-Am (blue), and Ni-NTA-Am (teal) are mixed to form a coacervate droplet in which His-tagged proteins are loaded. Upon addition of the terpolymer, the droplets are stabilized. These stabilized coacervates can be loaded with functional cargo to allow for an enhanced enzymatic activity or protease-mediated release.

sfGFP-His was sequestered into the coacervate core to give a local concentration of 40 μM (Fig. 2c and Supplementary Fig. 6). This concentration effect yields a ~150 fold increase in local concentration, highlighting the unique ability to specifically localize and concentrate functional cargoes, that otherwise cannot be incorporated without extensive modification, via this targeted $Ni^{2+}$-NTA/His interaction.

**Characterization of the protein-loaded coacervates**. To demonstrate the ability of the controlled uptake mechanism to sequester proteins irrespective of their surface charge, two additional His-tagged GFP variants with different surface charges were explored[36,39]. Both the negatively (−30GFP-His) and positively (+36GFP-His) supercharged variants were recombinantly expressed and obtained in high purity (Fig. 3a and Supplementary Fig. 10). Additionally, for each variant, the His-tag was removed by TEV protease (Supplementary Fig. 11). Without the His-tag, uptake was dominated by electrostatics, which is consistent with previously reported results[32], but by utilizing our programmed uptake mechanism, an increase of fluorescence was observed for all His-tag variants inside the coacervate core (Fig. 3b). In the case of +36GFP and sfGFP, a clear distinction in loading was observed in the presence of the targeted $Ni^{2+}$-NTA/His interaction, while for −30GFP this effect was not as pronounced as its interactions were still dominated by favorable electrostatics. The punctate structures present inside the protocells containing +36GFP(-His) are protein aggregates resulting from its instability in a low salt buffer (Supplementary Fig. 13). Next, we investigated the diffusivity of encapsulated cargoes, to ensure that our programmed uptake strategy does not eliminate cargo mobility. This is an important parameter, because for many proteins function is dependent on their spatiotemporal organization, for example in signaling cascades and multistep catalytic processes[40–42]. For other coacervate-based cell mimics, a wide range of apparent diffusion constants ($D_{App}$) for short RNA chains have been reported, 0.0002–1.9 μm² s⁻¹, and have been shown to be mainly dependent on the polyelectrolyte composition[17,43,44]. To

determine the role of the His-tag and charge in our system, the diffusion of sfGFP-His, −30GFP-His, and −30GFP was investigated with fluorescence recovery after photobleaching (FRAP). Due to the aggregated state of loaded +36GFP, it was not included for these experiments. The diffusivity of −30GFP provided a benchmark for the His-tagged proteins since it only has electrostatic interactions with the coacervate core. Both His-tagged proteins showed full recovery at similar rates, however, the diffusion rate of −30GFP was clearly faster compared to the other two His-tagged proteins (Fig. 3c). By plotting the fluorescence recovery against time, the data could be fitted with an exponential decay function to calculate $D_{app}$ (Fig. 3d and Supplementary Fig. 12). For the −30GFP, $D_{App}$ was found to be $0.368 \pm 0.017$ μm² s⁻¹, the sfGFP-His and −30GFP-His showed slower diffusion rates, $D_{app} = 0.076 \pm 0.002$ μm² s⁻¹, and $D_{app} = 0.038 \pm 0.001$ μm² s⁻¹, respectively. The similarity in diffusion rates for both His-tagged proteins indicates that the diffusion is mainly dominated by the His-tag and $Ni^{2+}$/NTA interaction rather than by the charged nature of the coacervates. Despite the decreased diffusivity, the His-tagged proteins are still able to diffuse through the core and are not immobilized, a prerequisite for any sequestered enzymes to retain activity upon loading.

**Enhanced enzymatic activity inside coacervates**. An interesting phenomenon seen in living cells is the localization of co-dependent molecules. This not only leads to high selectivity but also to high local concentrations, which can influence the rate of enzymatic reactions[24–26,45,46]. To explore the capabilities of our controlled macromolecular uptake mechanism, the conversion of L-tryptophan (L-Trp) to indigo by a synthetic, two-enzyme cascade was selected[47]. Tryptophan anhydrase (TnaA)[48] is responsible for the conversion of L-Trp to indole with pyridoxal-5-phosphate (PLP) as a cofactor. The indole is then oxidized by flavin-containing monooxygenase (FMO)[49], consuming nicotinamide adenine dinucleotide phosphate (NADPH) in the process (Fig. 4a). This cascade was selected for its simple readout and the fact that as a synthetic cascade, these enzymes do not have a

**a**

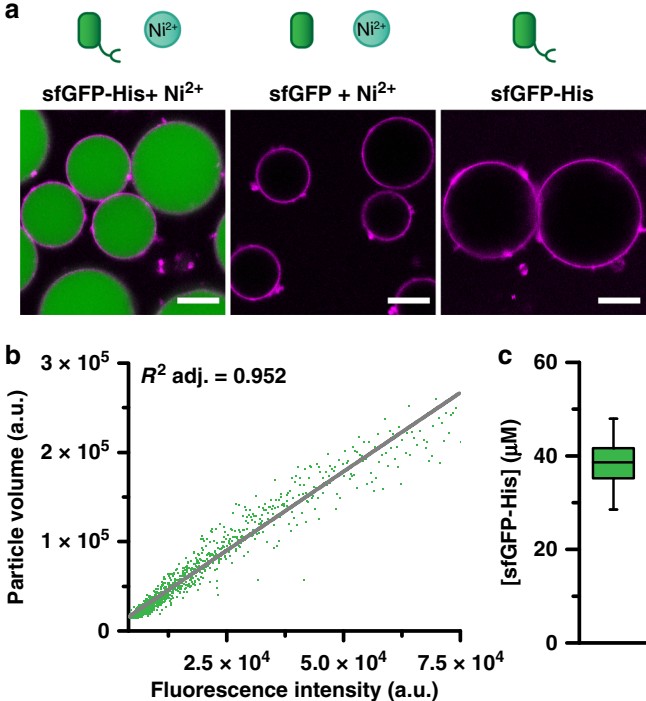

**Fig. 2 Controlled, homogeneous uptake of sfGFP-His into NTA-functionalized protocells. a** False-colored confocal micrographs of protocells show the specific uptake of sfGFP-His (green) by Ni-NTA-Am. The membrane is stained with Nile Red (purple), scale bar: 10 μm. Uncropped images are to be found in Supplementary Fig. 7. **b** Flow cytometry analysis of sfGFP-His loaded protocells. For clarity of the graph, a 10th of all data points collected is shown. Gating parameters and population statistics are reported in Supplementary Fig. 8. **c** Box plot analysis of the measured concentration of sfGFP-His measured inside the coacervate interior with confocal microscopy. Reference curve Supplementary Fig. 9. $n = 69$.

natural affinity for each other and thus were expected to perform better when brought together in a confined environment. In addition, due to their high surface lysine residue content, even gentle modifications of these residues (for example, with Sulfo-Cy5-NHS (Fig. 4c and Supplementary Figs. 18 and 19)) destabilize protein folding and decrease enzyme activity. As such, succinylation is not a viable option for their uptake into coacervate-based protocells, making this system a perfect candidate for $Ni^{2+}$-NTA/His driven uptake. Both His-tagged enzymes were expressed recombinantly and obtained in high purity (Fig. 4b and Supplementary Fig. 16). To enable visualization and quantification of enzyme uptake, both were functionalized with Sulfo-Cy5-NHS, resulting in a ~0.5 degree of labeling for both. The larger size of the enzymes, a 220 kDa tetramer, and 111 kDa dimer, for TnaA and FMO, respectively, did not hamper efficient sequestration as shown with confocal microscopy (Fig. 4c). Moreover, to fully show control over the uptake, the enzymes were added in a 1:2 ratio of TnaA:FMO. In order to quantify the absolute loading efficiency of each enzyme, two separate populations were prepared: Sulfo-Cy5-NHS-labeled TnaA with unlabeled FMO, and Sulfo-Cy5-NHS-labeled FMO with unlabeled TnaA. Analysis of the fluorescence intensity inside the coacervate core showed a ~2× higher intensity from FMO compared to TnaA (Fig. 4d). As an initial test, the cascade activity was confirmed with unlabeled enzyme-loaded coacervates in an overnight reaction at 30 °C, after which the coacervates were collected and dissolved in DMSO. It became immediately apparent that indigo

was produced due to the appearance of a blue color. When either the substrate mix or the enzymes were omitted, no blue color was observed (Supplementary Fig. 21). To study the effect of programmed co-localization on this reaction, NADPH consumption and indoxyl production were followed over time via absorption and fluorescence spectroscopy respectively. Within ~3.5 h all the NADPH was consumed by the enzymes sequestered in the coacervates (orange), compared to >12 h for the same number of enzymes in solution (gray); a clear increase in reaction rate was observed (Fig. 4e). By decreasing the total amount of enzyme for the 0.5:1 ratio by half, the reaction also takes twice as long, from ~3.5 to ~7 h. Meanwhile, in solution, a much lower rate is observed (Fig. 4f). Moreover, for both concentrations of enzyme, the indoxyl fluorescence intensity did not reach the same level inside the coacervates as in solution, showcasing the faster conversion of the intermediate (Supplementary Fig. 22). The potential of programmed cargo uptake can be demonstrated by investigating different ratios of loaded enzymes. Over a wide range of TnaA:FMO ratios, the overall rate stays the same with a constant amount of FMO (Supplementary Fig. 23), confirming that FMO is the rate-limiting enzyme. In solution, however, a clear distinction between the different ratios can be made, where there is a clear dependency on the amount of TnaA present for the reaction rate (gray) (Supplementary Fig. 23). Only when the system is pushed by a large excess of TnaA (2:1 TnaA:FMO), the rate in solution becomes similar to that in the coacervate system. This type of analysis would not be possible without a robust programmable uptake strategy. Interestingly, in the absence of $Ni^{2+}$, the time for complete NADPH consumption was similar to Fig. 4e (Supplementary Fig. 24). This was unexpected as TnaA has a similar theoretical isoelectric point to GFP (6.19 and 6.04, respectively) and thus should be excluded from the coacervate core. Confocal microscopy indicated non-specific TnaA adsorption on the periphery of the protocells, accomplishing co-localization in the same microenvironment but without the control over the enzyme stoichiometry (Supplementary Fig. 25). This illustrates the need for a generally applicable, bio-orthogonal uptake strategy, which delivers control over both the local concentration and co-localization of the enzymes involved. These data represent an important progression towards the study of complex enzymatic cascades in confined, discrete, cell-mimetic environments.

**TEV protease-mediated release.** As a final proof of the functional diversity that can be accomplished with this protocell platform, a unique cargo release mechanism was designed, which utilizes both electrostatic interactions and enzymatic control. As established earlier in Fig. 3b, sfGFP is not taken up into the coacervate core unless it is His-tagged. Thus, if the His-tag can be cleaved from the sfGFP-His cargo, excretion of the protein from the protocell system would be expected (Fig. 5a). To engineer this release, a commercially available, His-tagged TEV protease was selected, which specifically cleaves the amino acid sequence between sfGFP and the hexahistidine tag in our constructs. After loading both the sfGFP-His and TEV protease into the protocells, a clear drop in fluorescence intensity was observed inside the coacervate core after one hour of incubation (Fig. 5b). This release was not observed when TEV protease was omitted. Similarly, for +36GFP release was also observed for when incubated with the TEV protease (Supplementary Fig. 28), but again was not studied in detail due to protein aggregation. As an additional control, −30GFP-His was loaded, which resulted in no observable release, a consequence of the strong electrostatic interactions between the coacervate core and the negatively charged protein, negating the effect of the His-tag. In order to

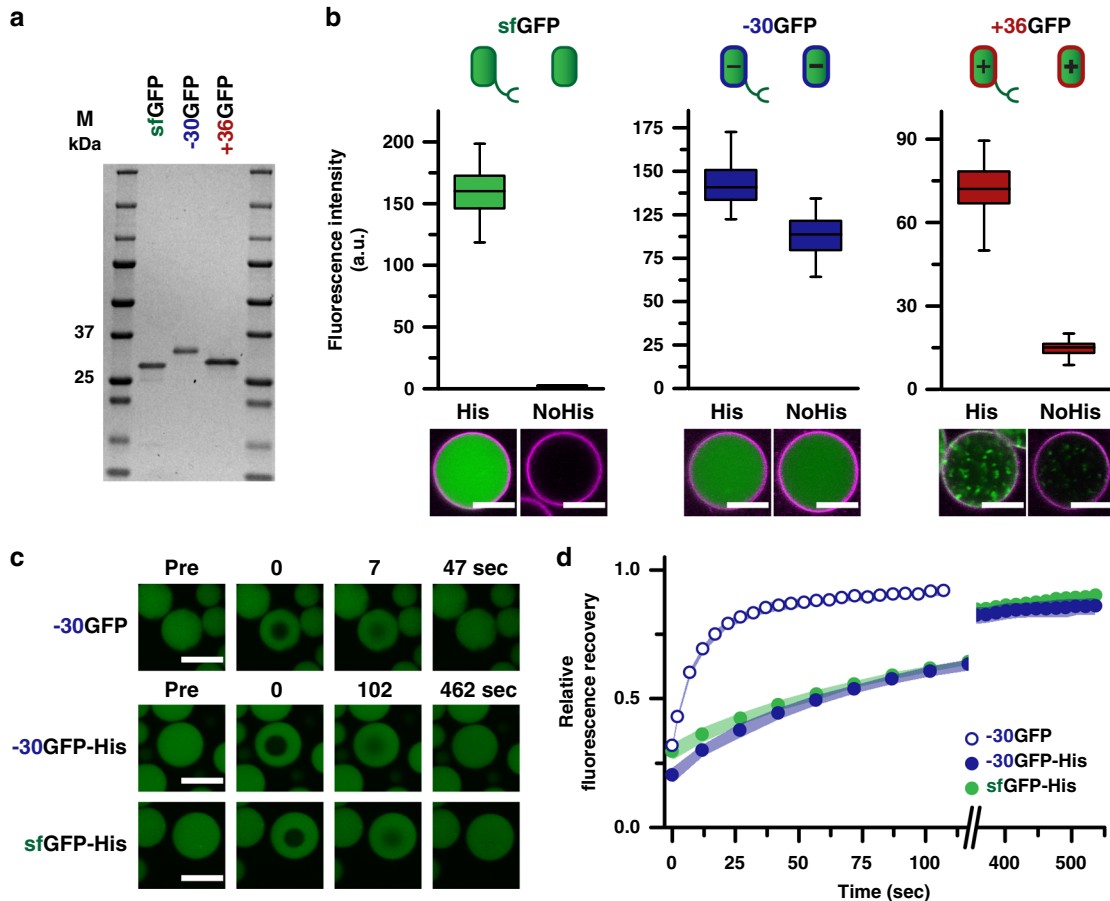

**Fig. 3 Detailed characterization of the Ni²⁺-NTA/His uptake mechanism. a** Analysis of the purified GFP variants by SDS-PAGE analysis. Left to right: marker (M), purified sfGFP-His, purified −30GFP-His, +36GFP-His, M. For all three GFP variants, a prominent band is observed at the expected molecular weight. **b** Analysis of coacervate populations containing either the His-tagged or No His variant of each supercharged protein. Charge is indicated by color; green, neutral; blue, negative; and red, positive. For all cases, n > 60. The membrane is stained with Nile Red, scale bar: 10 μm. Uncropped images are available in Supplementary Fig. 12c, d. FRAP analyses of protein-loaded coacervate protocells. **c** Confocal micrographs at different time points during the recovery experiments. Scale bar: 10 μm. **d** Analysis of the fluorescence recovery over time. Open blue circles, −30GFP without His-tag; closed blue circles, −30GFP-His; and green circles, sfGFP-His. Shaded areas represent the standard deviation of three individual measurements. The unbroken graph and data fits are available in Supplementary Figs. 14 and 15.

confirm the in-situ removal of the His-tag, the diffusivity of the cargo in the protocell core was determined via FRAP experiments (Fig. 5c and Supplementary Fig. 29). For sfGFP-His, a slow recovery was observed with a $D_{app}$ of 0.092 μm² s⁻¹, which is in line with the $D_{app}$ of sfGFP-His reported in Fig. 3d. This indicates that FRAP predominantly proceeds with the His-tagged protein, as the sfGFP-His cleaved by the TEV protease has been excluded from the coacervate core. In the case of −30GFP-His, a fast recovery was seen with a $D_{app}$ of 0.279 μm² s⁻¹, which is similar to the observed diffusion speed of −30GFP without His-tag (Fig. 3d), confirming that TEV protease has cleaved between the protein and His-tag, and the −30GFP remains due to electrostatics. In order to gain an insight into the kinetics of protease-mediated cargo excretion, confocal microscopy was utilized to measure the amount of protein left inside the coacervate core over time (Fig. 5d). Once again a clear difference was observed when TEV protease was excluded, and the release of sfGFP proceeded in a gradual manner, as expected for an enzyme-mediated process. Moreover, by adding different amounts of TEV protease the rate of release can be tuned (Supplementary Fig. 31). However, after 60 min the rate of all concentrations is similar, indicating that the system is diffusion-limited. This can be explained by the low apparent diffusion coefficient for His-tag loaded proteins in this system. Despite the slower diffusivity, the system

approximates full release after 2 h (Supplementary Fig. 32). The ability to achieve enzyme-mediated protein secretion opens up many possibilities for the release of biologically active proteins, such as cytokines and growth factors. Furthermore, the concept could be easily extended toward the light or chemically triggered release as well.

## Discussion

In this work, we have designed a methodology for the introduction of biologically inspired function in protocells. In the field of synthetic cells, advances in functional capabilities are achieved by incorporating increasingly complex arrays of both synthetic and natural macromolecular species. However, using current encapsulation techniques, there are several challenges that have yet to be surmounted. The first of these, specific to statistical encapsulation, is a typically low encapsulation efficiency, which limits the amount of included cargo, impacting functional performance and necessitating purification procedures to remove non-encapsulated cargoes. Secondly, there is either poor control over the final composition of encapsulated species (statistical) or non-biased uptake (membrane-free protocells), which are particularly undesirable when attempting to efficiently encapsulate a large number of different species. These challenges clearly stand

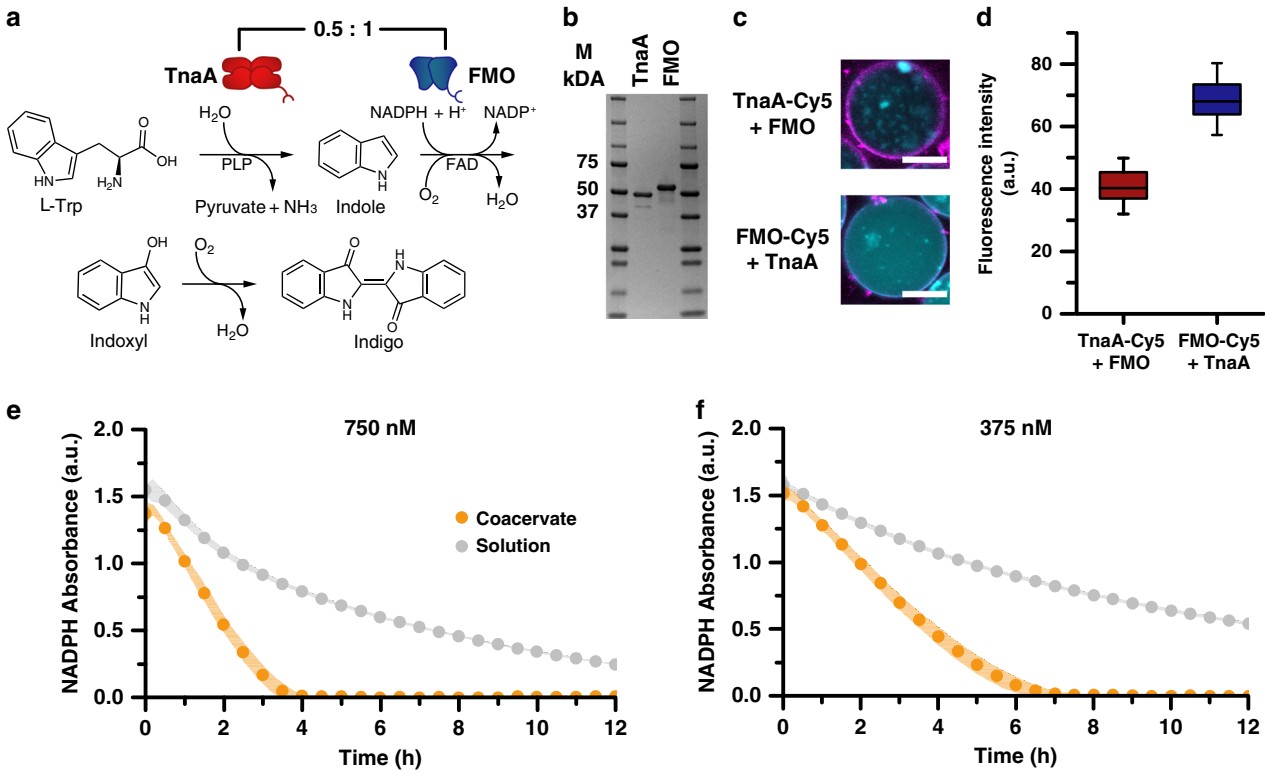

**Fig. 4 Increased enzymatic activity by controlled loading of a synthetic two-enzyme cascade. a** Schematic overview of the reaction scheme to produce Indigo from L-Trp by TnaA and FMO. **b** Analysis of the purified enzyme variants by SDS-PAGE gel electrophoresis. Left to right: marker (M), TnaA, FMO, M. **c** Confocal micrographs of coacervates loaded with 100 nM Sulfo-Cy5-NHS-labeled TnaA + 200 nM unlabeled FMO (top) or 200 nM FMO + 100 nM unlabeled TnaA (bottom). Membrane stained with Nile red (purple), scale bar: 10 μm. Uncropped images are to be found in Supplementary Fig. 18. **d** Box plot analysis of the loading of fluorescently labeled enzymes inside the coacervates. For all cases, $n > 50$. **e, f** NADPH consumption measured by the absorbance at 340 nm. 1 mM L-Trp and 0.5 mM NADPH was added prior to the measurement. Orange circles represent enzymes inside the coacervates, gray circles show enzymes in solution. Three individual batches of coacervates were made for each condition, the standard deviation of three individual measurements is represented by the shaded area. For clarity of the graph, only 1/3rd of all collected data points are shown. **e** 250 nM TnaA + 500 nM FMO. **f** 125 nM TnaA + 250 nM FMO.

in the way of creating bottom-up protocells with a degree of cargo complexity on par with nature and have been overcome with the development of the system described herein.

The inclusion of the biocompatible $Ni^{2+}$-NTA/His-tag binding motif within discrete, stable protocells has resulted in control over the concentration and stoichiometry of incorporated cargo. Significantly, this method is not reliant on statistical encapsulation efficiencies, microfluidic methodologies, or the intrinsic interactions between cargo and the protocell platform. These benefits have been highlighted by the incorporation of the synthetic indigo producing cascade, where enhanced activity was obtained using His-tagged, aggregation-prone enzymes. However, such high levels of local enzyme concentration expose parameters that need to be accounted for in reconstituted synthetic cascades. There is still a limited understanding of substrate/cofactor (e.g., amino acids, nucleotides, NADPH, etc.) localization and availability in this system, which were until now not necessary to consider. This substrate bottleneck has also been observed in synthetic enzyme-loaded protein-based compartments, where high local catalyst concentrations do not lead to higher rates of substrate turnover[39,50,51]. In future investigations, this limitation could be overcome by taking inspiration from nature, such as employing active uptake or carrier molecules.

One of the key strengths of this approach is its general applicability, as the utilization of the $Ni^{2+}$-NTA/His-tag interaction is already widespread, being one of the most commonly used protein purification methods. This results in near-limitless

possibilities for the controlled sequestration of protein-based cargoes, which can be easily obtained from commercial sources or recombinantly expressed and purified via existing protocols. Moreover, the orthogonality of the Ni-NTA/His-tag approach could be further expanded by exploring additional site-specific protein binding interactions, such as SNAP-tag, maltose-binding protein, and biotin-streptavidin. Many of these are used for specific and controlled binding of recombinantly expressed proteins in either purification strategies, coupling reactions, or assays[52–54]. Covalent attachment of the required binding motifs to either the terpolymer or the amyloses would enable uptake mechanisms that function independently of each other, thus expanding the toolbox of orthogonal and controlled loading of biomacromolecules into the coacervate core.

Finally, careful balancing of accretive ($Ni^{2+}$-NTA/His-tag) and secretive (electrostatic repulsion) forces within the coacervate core via TEV protease not only highlights the fine level of control in this system but demonstrates its flexibility and use as a protein secretion platform. While currently cargo release is limited to positively charged cargo, this could be broadened through further engineering of the intrinsic molecular interactions between the cargo and protocell scaffold. Furthermore, balancing the interplay between accretive and secretive forces could provide control over the rate of release, and the introduction of a trigger mechanism would enable regulation over the spatiotemporal dynamics of local macromolecule concentrations. This technology has many exciting applications, and could therefore be adopted for the

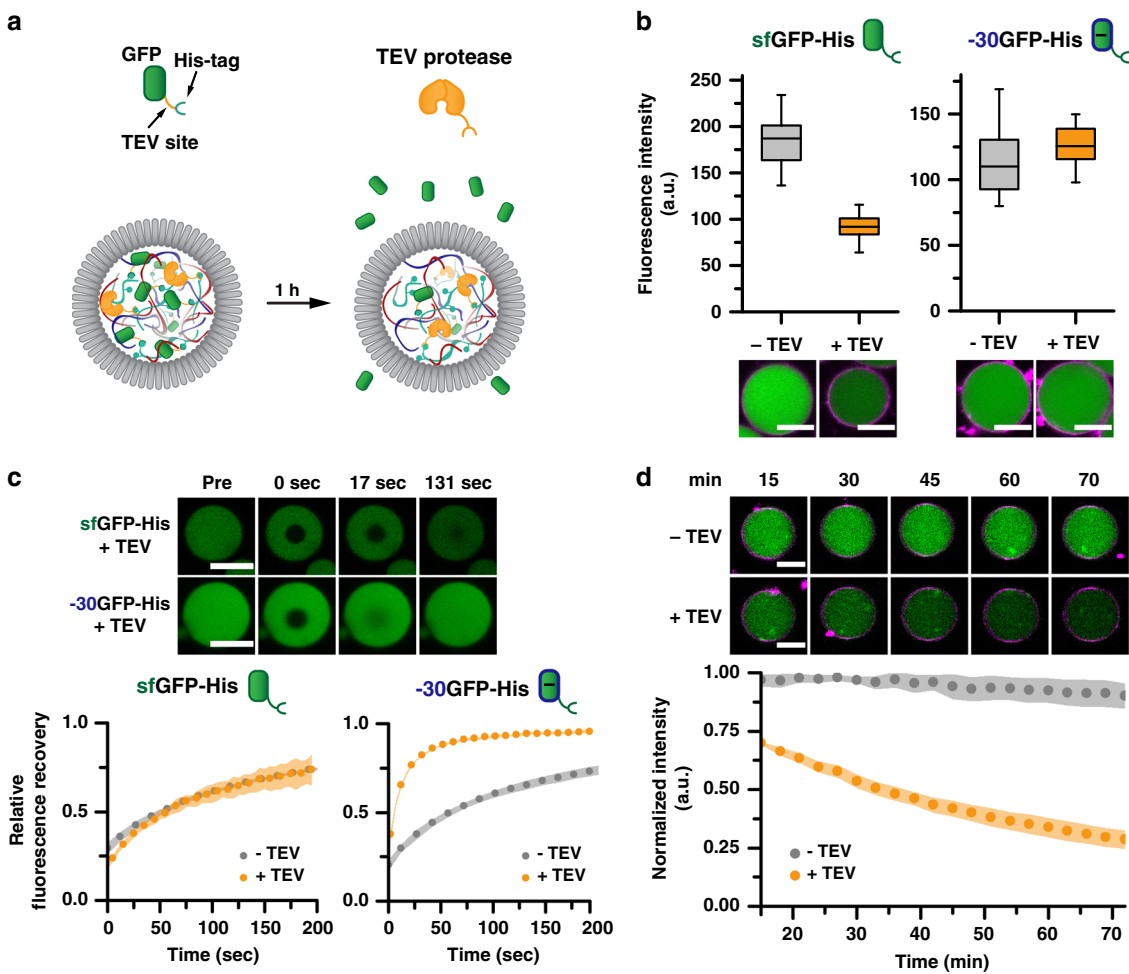

**Fig. 5 Protein excretion induced by TEV protease-mediated His-tag removal. a** Schematic overview of the GFP construct design and the TEV protease-mediated release. **b** Box plot analysis of the fluorescence intensity inside the coacervate droplet incubated with (orange) and without (gray) TEV protease for both 100 nM of sfGFP (left) and −30GFP (right). $n > 30$. Membrane stained with Nile red (purple), scale bar: 10 μm. Uncropped images (Supplementary Fig. 27). **c** FRAP analysis of both the sfGFP and −30GFP after 1 h of incubation with TEV protease. Top: confocal images of the FRAP experiment. Scale bar: 10 μm. Bottom: Fluorescence recovery plotted over time. Left: sfGFP with TEV (orange), sfGFP-His without TEV (gray). Right: sfGFP with TEV (orange), −30GFP without TEV (gray). The standard deviation of three individual measurements is represented by the shaded area. Data fits available in Supplementary Fig. 29. **d** Release profile of 250 nM sfGFP determined by confocal microscopy. Top: Time series of representative protocells. Membrane stained with Nile Red (purple), scale bar: 10 μm. Uncropped images (Supplementary Fig. 30) Bottom: Normalized fluorescence intensity of $n = 20$ protocells extracted from confocal time series. The standard deviation is represented by the shaded area.

delivery of growth factors in tissue engineering, targeted protein-drug release, and on a more fundamental level, for the modeling of protein-based signaling pathways in a controlled environment.

In summary, we have demonstrated an elegant, robust, and general method for the controlled loading and release of a broad range of His-tagged recombinant proteins into coacervate-based protocells via $Ni^{2+}$-NTA/His interactions. The efficient and programmed uptake of near-native macromolecular cargo, modified only with a His-tag, into discrete protocells opens up an enormous range of possibilities for exploring protein-based biological processes, from enzymatic cascades to signaling pathways. This work represents a significant step forwards for the field of bottom-up synthetic cells and provides a robust, adaptable, and accessible foundation for the creation of increasingly complex cell-mimetic microenvironments.

## Methods

**DNA molecular biology and cloning**. All DNA was ordered through Integrated DNA Technologies (IDT). The sequence of each protein was derived from either Lawrence et al.[36] for the GFP's, and Giessen et al.[47] for the enzymes TnaA (Tryptophanase) and FMO (Flavin-containing monooxygenase). The constructs

were optimized using the IDT Codon Optimization Tool for *Escherichia coli* (*E. coli*) based on the amino acid sequence. The vector pET-28a and the gBlocks were both digested using the restriction enzymes *Nco*I and *Xho*I (New England Biolabs). After ligation, the constructs were verified using Sanger sequencing (BaseClear). Constructs were transformed into BL21(DE3) bacterial cells (Novagen).

**Protein expression and purification**. For all proteins, 600 mL 2YT medium supplemented with kanamycin (100 μg mL$^{-1}$) was inoculated using an overnight culture grown at 37 °C, 200 rpm to an optical density (OD600) of 0.6, and then induced with isopropyl β-D-1-thiogalactopyranoside (IPTG) (0.1 mM final concentration). The proteins were expressed at 18 °C, 150 rpm for 18 h. The cells were harvested by centrifugation (2700×*g*, 15 min, 4 °C) and the pellet was either lysed immediately or flash-frozen in liquid N$_2$ and stored at −20 °C.

For the GFP variants, the purification protocol was based on previous reports[36,39]. The cell pellets were resuspended in lysis buffer (50 mM NaH$_2$PO$_4$, 300 mM NaCl, 10 mM imidazole, pH 8, and for the +36GFP variant, 1 M NaCl was used in addition) and lysed using an EmulsiFlexC3 High-Pressure homogenizer (Avestin) at 15,000 psi for three executive rounds. For +36GFP, 10 mg mL$^{-1}$ DNaseI (PanReac AppliChem) and 5 mg mL$^{-1}$ RNase A (ThermoFisher) were added 30 min before lysis. Cell debris and insoluble proteins were removed by centrifugation (15,000×*g*, 10 min, 4 °C). The His-tagged proteins were purified from the soluble lysate using Ni-NTA affinity chromatography (His-Bind Resin, Novagen). The lysates were incubated with the His-bind resin for 1.5 h at 4 °C on a shaking table, prior to loading onto an empty gravity flow column (Bio-Rad). The

resin was washed twice with wash buffer (50 mM NaH$_2$PO$_4$, 300 mM NaCl, 20 mM imidazole, pH 8; again for the +36GFP variant, 1 M NaCl was used). The His-tagged proteins were eluted from the resin using elution buffer (50 mM NaH$_2$PO$_4$, 300 mM NaCl, 250 mM imidazole, pH 8; for the +36GFP variant, 1 M NaCl was used).

In case of TnaA and FMO proteins, the purification was based on Giessen et al.[47]. The cell pellets were resuspended in Buffer A (50 mM Tris, pH 8, 300 mM NaCl, 5% glycerol, 30 mM imidazole) and lysed using an EmulsiFlexC3 High-Pressure homogenizer (Avestin) at 15,000 psi for three executive rounds. The cell debris and insoluble proteins were separated by centrifugation at 15,000 × g, 4 °C for 10 min. The supernatant flowed over a Buffer A equilibrated HisTrap FF 5 mL Column (GE Healthcare) with a rate of 2 mL min$^{-1}$ controlled by a peristaltic pump. After loading of the lysate, the column was washed with 4 column volumes of buffer A. The His-tagged proteins were eluted in two-column volumes of Buffer B (50 mM Tris, pH 8, 300 mM NaCl, 5% glycerol, 250 mM imidazole).

Extensive dialysis was performed to remove the imidazole using a 10 kDa molecular weight cut off (MWCO) membrane (Millipore), into phosphate-buffered saline (PBS), pH 7.4 (for the sfGFP, −30GFP, TnaA and FMO proteins) or 50 mM NaPi, 600 mM NaCl, pH: 7.5 (storage buffer for the +36GFP protein). The samples were concentrated using 3 kDa MWCO Amicon Ultra Filters (Millipore). For further purification, the samples were passed through a HiLoad Superdex 26/600 200 pg preparative column (GE Healthcare) connected to an ÄKTApurifier FPLC (GE Healthcare) at a flow rate of 1 mL min$^{-1}$, and the absorbances at 280 nm and 488 nm (GFP variants only) were monitored. The eluted fractions were analyzed using SDS-PAGE (4–20% Mini-PROTEAN TGX Precast Protein Gel, Bio-Rad) and the purest fractions were pooled and concentrated. For the FMO enzyme specifically, only the fractions from peaks 1 and 2 were pooled (Supplementary Fig. 17). The other two peaks correspond to the protein without FAD, visible by the absence of a yellowish color at high μM concentration, and are not functional. Protein concentrations were calculated using the 280 nm absorbance determined by the ND-1000 spectrophotometer (ThermoScientific) and theoretical extinction coefficients of 20,400 M$^{-1}$ cm$^{-1}$ (for the GFP variants), 49,740 M$^{-1}$ cm$^{-1}$ (TnaA protein), and 118720 M$^{-1}$ cm$^{-1}$ (FMO protein), as determined by the online ProtParam tool (ExPASy). The samples were aliquoted for single-use, flash-frozen in liquid N$_2$, and stored at −80 °C. The identity of the protein samples was confirmed using liquid chromatography quadrupole time of flight mass spectrometry (LC–MS Q-TOF) (Supplementary Fig. 10 for the GFP proteins, Supplementary Fig. 16 for the enzymes).

**Removal of the His-tag.** To remove the histidine-tag from the GFP variants, 1 mg of the target protein was incubated overnight at 4 °C with 200 U of TEV-H$_6$ protease (Protean) and 1 μM TCEP. The cleaved protein was extracted by flowing the solution over a gravity flow Ni-NTA affinity column (His-Bind Resin, Novagen). The flow-through containing proteins without the histidine-tag was collected and concentrated using Amicon Ultra Filters with 3 kDA MWCO (Millipore). The successful removal of the histidine-tag was confirmed by LC–MS Q-TOF (Supplementary Fig. 11).

**Synthesis of modified amyloses.** Both quaternized (Q-Am) and carboxymethylated (CM-Am) amylose were synthesized as was reported in previously published procedures. (Supplementary notes, Supplementary Fig. 1)[31]. For both Q-Am and Cm-Am, 12–16 kDa amylose (Carbosynth) was dissolved in aqueous NaOH. In the case of Q-Am, 3-chloro-2-hydroxypropyltrimethylammonium chloride solution (60 wt% in water) was added into a stirring reaction mixture in a dropwise fashion. Subsequently, the reaction was left overnight at 35 °C. For CM-Am, chloroacetic acid was added to a stirring mixture and left for 2 h at 70 °C. Both modified amyloses were purified by precipitation into cold ethanol. Prior to lyophilization, the products were dialyzed extensively against ultrapure water.

Nitrilotriacetic acid-modified amylose (NTA-Am) was prepared via EDC/NHS activation of the CM-Am carboxylic acid (Supplementary Fig. 2), followed by amide bond formation with an amine-functionalized NTA. First, CM-Am (85 mg, 0.39 mmol eq.) was dissolved in 10 mM NaHPO$_4$ buffer (10 mL) adjusted to pH 6 with 1 M HCl. To this was added N-hydroxysuccinimide (67 mg, 0.58 mmol) and 1-ethyl-3-(3-dimethylaminopropyl)carbodiimide (222 mg, 1.16 mmol). The reaction mixture was then stirred for 2 h at room temperature. This mixture was subsequently concentrated using 3 kDa MWCO spin filters, diluted with 10 mM NaHPO$_4$ buffer (adjusted to pH 8), and concentrated again to remove reagents and change the pH of the reaction medium for the next step. This centrifuge/dilution cycle was repeated a further two times. For the conjugation of the NTA group, N$_\alpha$, N$_\alpha$-bis(carboxymethyl)-L-lysine hydrate (152 mg, 0.58 mmol) was first dissolved in 18 mL of 10 mM NaHPO$_4$ buffer (adjusted to pH 8) with 5% DMSO. To this solution was added the NHS-activated CM-Am, and the reaction mixture was left to stir at room temperature overnight. The reaction mixture was concentrated, dialyzed extensively against MilliQ water, and lyophilized to yield NTA-Am (160 mg, ca. 90%) with a degree of substitution of 0.12. $^1$H NMR (D$_2$O) characterization data are presented in Supplementary Fig. 5.

**Synthesis of terpolymer.** PEG-b-PCLgPTMC-b-PGA was prepared as described by previously published procedures (Supplementary notes, Supplementary Fig. 3)[31]. In short, the ring-opening polymerization of ε-caprolactone and trimethylene carbonate was initiated by the use of poly(ethylene glycol) monomethyl ether. The polymeric terminal alcohol was modified with Boc-L-Phe-OH via a Steglich ester-ification, which resulted in a primary amine after deprotection with TFA. In order to connect the final poly(L-glutamic acid) block, a ring-opening polymerization of N-carboxyanhydride γ-benzyl L-glutamate was performed. Subsequent hydrogenation yielded the PEG-b-PCLgPTMC-b-PGA polymer.

**Coacervate preparation, protein loading, and imaging.** Q-Am, CM-Am NTA-Am were dissolved separately in PBS at a concentration of 1 mg mL$^{-1}$. First 5 μL NTA-Am was added to 7.5 μM of NiSO$_4$ (final concentration) in a 1.5 mL tube shaking at 1500 rpm in a MixMate (Eppendorf). Consecutively, CM-Am, Q-Am, and NTA-Am were added to induce coacervation in a 2:0.8:0.2 mass ratio of Q-Am:CM-Am:NTA-Am. While different ratios of modified amyloses can be used, the ratio of 2:1 Q:CM, corresponding to approximately a 3:1 charge ratio due to differing degrees of substitution, has been found to be the most stable[31]. After 30 s, 50–750 nM of protein was added to the shaking solution (see figure captions or methods for details of a specific experiment). To achieve stabilized particles with ~20–25 μm in diameter 5 μL terpolymer (50 mg mL$^{-1}$ in PEG350 containing 15 μM Nile Red) was added after 6 min and shaken for another 5–10 s. For analysis, 150 μL of each sample was loaded on a μ-side 8 well glass bottom (Ibidi). Unless state otherwise, the samples were analyzed with a Zeiss LSM510 META NLO equipped with a C-Apochromat, 63 × 1.2 UV-VIS-IR water objective, 488/545/633 nm Dichroic, PMT detector. For imaging of the GFP, an Argon laser set at 488 nm and a BP filter of 500–530 nm were used. Nile red staining was visualized using a 545 nm He/Ne laser and a BP filter of 565-615 nm. For each wavelength, the pinhole was set to 1 airy unit. The transmission and detector gain were optimized for each different fluorescent protein/fluorophore to utilize the maximum amount of gray values of the detector unless stated otherwise. Images of 2048 × 2048 pixels were acquired with a pixel dwell of 1.6 μs.

**Single-particle image analysis.** All images were analyzed with Fiji (ImageJ). In order to determine the fluorescence intensity, a two-color image of the interior and the membrane of the coacervate was required. Only standard ImageJ functions were used for the analysis. In the membrane image the polymer layer was selected by thresholding the image and converting it into a binary image. To determine the Region of Interest (ROI's), the coacervate droplets, watershed function was used followed by analyzing particles. Alternatively, Hough Circle Transform (UCB Vision Sciences) was used to determine the membrane section of each droplet when multiple time points were required. Since both algorithms are not fully selective, the ROI's corresponding with coacervates was picked by hand. Next, the ROI's were applied to the coacervate interior image and the intensity was determined for each selected particle.

**Flow cytometry analysis.** Coacervates loaded with 250 nM sfGFP-His were analyzed with an Aria III (BD Biosciences) FACS equipped with a 70 μm nozzle, 488 nm laser, and a 585 ± 7.5 nm bandpass filter. Single coacervates were selected based on the forward scatter versus side scatter (Supplementary Fig. 8). The fluorescence intensity of 1.000 individual droplets was collected. Any data point that was at the maximum value of the detector was removed from the data set.

**Measuring local GFP concentration and calibration curve.** Preparation and imaging of the coacervate droplets were performed as described above. The calibration slide was prepared by placing a Press-to-Seal™ Silicone Isolator with Adhesive, eight wells, 9 mm diameter, 0.5 mm deep (Invitrogen) on isopropanol cleaned super frost micro slide (VWR). Following, 20 μL sfGFP-His protein ranging from 1 to 50 μM was loaded onto the slide and sealed with a glass coverslip (VWR). The calibration slide was imaged with the exact same settings as the sfGFP-His loaded coacervate sample. The fluorescence intensity of each concentration was determined using Fiji.

**FRAP (fluorescence recovery after photobleaching).** Coacervate droplets were prepared as described above and loaded with 250 nM of a GFP variant. 100 μL of each sample was transferred on a μ-side 8 well glass bottom (Ibidi). FRAP experiments were performed with the bleaching interface available in Zen 2009 (Zeiss). For imaging, the same settings were used as described above. An initial image was acquired in order to define the region of interest (ROI), 5–10 μm in diameter, within a coacervate. Following, three images of 1024 × 1024 with a pixel dell of 0.8 μs were acquired prior to the bleaching. Subsequently, the ROI was bleached for 10 iterations with a two-photon laser (Chameleon, Coherent) set at 810 nm, 50% laser power. The recovery was monitored with a 5 s interval, for a total of 33 images. The intensities of the bleached ROI, reference area, a nearby coacervate that was not bleached, and background were extracted from the images with FIJI. Data were normalized by removing the background intensity and dividing by the intensity of the reference area, as described by Jia et al.[44]. A first-order exponential equation was fitted using Origin 2019 (OriginLab) from which the half-life and $D_{app}$ were calculated as reported by Poudyal et al.[17].

**Labeling of the enzymes with sulfo-Cy5-NHS**. A few grains of sulfo-Cy5-NHS (Lumiprobe) were dissolved in ddH$_2$O. The concentration was determined using the absorption at 640 nm, $\varepsilon = 271{,}000$ cm$^{-1}$ M$^{-1}$ using an ND-1000 spectro-photometer (ThermoScientific). The reagent was added in an equimolar ratio to 60 μM of TnaA or FMO protein (~1:30 ratio of NHS to lysine residues for both enzymes) in PBS and left for 1 h at 4 °C. Unreacted dye was removed using a PD-25 spin trap column (GE Healthcare). The average labeling per protein was measured using the absorption of the dye at 640 nm and at 280 nm for the protein, TnaA-Cy5 degree of labeling: 0.48, FMO-Cy5 degree of labeling: 0.56. For the experiments regarding the uptake without Ni$^{2+}$ present, the degree of labeling was TnaA-Cy5: 0.52, FMO-Cy5: 0.32. The loading and imaging of these labeled proteins were performed as described above. For excitation of the Cy5 dye a 633 nm He/Ne laser was used. The brightfield images were made with a Zeiss Axio Observer D1 equipped with an LD Plan-Neofluar 40×/0.6 Corr, Halogen lamp, and an AxioCamMR3.

**Enzymatic activity inside coacervates**. Coacervates were formed as described above. Experimental settings were based on Giessen et al.[47].

*Enzymatic assay*. 150 μL of enzyme-loaded coacervates were incubated overnight at 30 °C with 0.5 mM l-Trp, 5 μM PLP and 0.5 mM NADPH. The next day, the samples were centrifuged for 1 min at 12,000 × g and the supernatant was removed. In order to dissolve and visualize the produced indigo, 20 μL of DMSO was added to the coacervate fraction.

**Enzymatic activity measurements**. Technical triplicates of enzyme-loaded coacervates, containing different amounts of enzyme were made described as above. For each condition, an enzyme mastermix was created from which each sample was made, with the following final concentrations: 125:250, 125:500, 250:500, 1000:500 nM of TnaA:FMO, respectively. In all cases, TnaA and FMO were preincubated with PLP, 5× excess compared to the TnaA concentration, at room temperature for at least 15 min. After formation, the samples were left for 5–10 min on the bench. Following, the sample was transferred to a Non-Binding black microplate 384 well with a transparent bottom (Greiner Bio-One) and a substrate mastermix containing 1 mM l-Trp and 0.5 mM NADPH was added. NADPH consumption was monitored for over 12 h, and a measurement was taken every 10 min using the absorption at 340 ± 20 nm as well as the indoxyl fluorescence (ex. 375 ± 20 nm, em. 470 ± 20 nm) at 30 °C using a Spark 10 M plate reader (Tecan). To prevent evaporation the plates were sealed with EASYseal™ transparent sealing film (Greiner Bio-One). In the case that the endpoint was not yet reached at the end of the 12 h, the plate was moved to a 30 °C incubator and measured at least 3 data points again after another 6 h, 750 nM of the enzyme, or 24 h, 375 nM of the enzyme. All data were normalized to their endpoint, flat line in absorbance, indicating that all NADPH was consumed. Without PLP or at temperatures below RT, the TnaA enzyme is not able to form its active tetrameric assembly, which results in a large variation between samples inside the coacervates (Supplementary Fig. 26).

**TEV protease-mediated release**. 100 nM of sfGFP-His, −30GFP-His or +36GFP, 5 U of TEV-H$_6$ protease (Protean) and 13 nM TCEP were loaded into the coacervates as described above. After 1 h of incubation at room temperature, the samples were analyzed with confocal microscopy as described earlier. For sfGFP and −30GFP, FRAP analysis was also performed on these samples.

*Time traces*. The samples for microscopy were prepared as described above containing 250 nM of sfGFP-His, 6.25 (0.5×), 12.5 (1×), 25 U (2×) of TEV-H6 protease and 16.5, 33, 66 nM TCEP, respectively; as a reference, 250 nM of sfGFP-His was used without any TCEP or TEV. The samples were analyzed with a Leica DMI8-CS equipped with an HC PL APO CS2 63×/1.20 water objective, 488/552 nm Dichroic, HyD detector. For imaging of the GFP, an OPSL set at 488 nm and a BP filter of 500–560 nm were used. Nile red staining was visualized using an OPSL of 554 nm and a BP filter of 560–662 nm. For each wavelength, the pinhole was set to 1 airy unit. The transmission and detector gain were optimized for each different fluorescent protein/fluorophore to utilize the maximum amount of gray values of the detector unless stated otherwise. Images of 1024 × 1024 pixels were acquired with a pixel dwell of 0.4 μs with a line averaging of 4. For each sample, 4–5 positions were analyzed every 3 min.

**Reporting summary**. Further information on research design is available in the Nature Research Reporting Summary linked to this article.

## Data availability

The data that support the findings of this study are available from the corresponding author upon reasonable request.

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

## Acknowledgements

The Dutch Ministry of Education, Culture, and Science (Gravitation Program 024.001.035) and the ERC Advanced Grant (Artisym 694120) are acknowledged for funding. Joost L. J. van Dongen and Iris A. Leijten - van de Gevel are thanked for the acquisition of the LC/MS Q-TOF data.

## Author contributions

W.J.A., A.F.M., and J.C.M.H. wrote the manuscript; W.J.A., A.F.M., N.A.Y., and J.C.M.H. designed the research; W.J.A., A.F.M., D.F.M.V., and M.H.M.E.v.S. performed the experiments. All authors reviewed the manuscript.

## Competing interests

The authors declare no competing interests.
