## [Peer Review File · Nature Communications]

Reviewers' comments:

Reviewer #1 (Remarks to the Author):

The manuscript by Wiggert et al. describes the formation of synthetic protocells that are capable of controlled loading and release of protein cargo. The protocells are based on coacervates that are formed by mixing of positively charged and negatively charged modified amyloses and then stabilized with a synthetic block copolymer coating. The inclusion of an NTA-modified amylose in the coacervate mixture enables the Ni-mediated sequestration of His-tagged proteins regardless of their surface charge, allowing the enhancement of a two-enzyme cascade reaction. Through the use of a cleavable His-tag, cargo proteins can also be released by enzymatic treatment.

Critique

The authors provide a very clear overview of the goals, current limitations and challenges in the area of synthetic protocells, which places this contribution into its broader context quite well. The specific issues with current protocell mimics and the way in which this work addresses them are plainly stated. Overall the manuscript is well-written, logical and easy to follow. The derived conclusions are supported by the data provided and the experimental work is comprehensive and of high quality in both execution and presentation. The general strategy reported seems promising for further development to increase the complexity and functionality of synthetic protocells. For these reasons, this manuscript should be of interest to the broad readership of Nature Communications and be ready for publication after some minor adjustments.

Specific comments

Charge ratio – the need for a 2:1 ratio of Q-Am to Cm-Am is unclear in this manuscript and should be explained briefly at the beginning of the results section. It seems that some residual positive charge is needed to drive interfacial assembly of the glutamate-containing terpolymer, but this could be stated explicitly. Confounding this issue are the different degrees of substitution for the Q-Am and Cm-Am, which are 0.88 and 0.4, respectively. Are the mixtures used for coacervate formation based on molar or charge ratios?

Figure 2b – the authors state there is a linear correlation between particle volume and fluorescence intensity. While the curve is approximately linear at lower values, it seems to deviate above 50000 a.u. revealing a plateau. For this reason, it would be worth acknowledging this in the main text. More crucially it would then be important to work in the linear range for subsequent experiments, specifically those quantifying different cargo loading and release (Figs. 3b, 4d, 5b).

Punctate structures are observed for some of the encapsulated proteins (e.g. GFP36+). The authors conclude that this is due to salt-induced aggregation. Are these aggregates present before incubation with coacervates, are they only observed within the protocells, and could they be mediated by free Cm-Am, terpolymer or NTA-Am chains? Although the FRAP experiments are not shown for GFP36+, the diffusion is presumably slowed, this seems to be missing from the discussion in lines 136-149 and should be addressed.

Quantification of enzyme co-encapsulation. The authors state that the ratio of the two enzymes can be tightly controlled, which is determined by microscopy. However, since both enzymes are labelled with Cy5, it is currently unclear how they could be distinguished when co-encapsulated. Can the authors also rationalise the aggregated structures of both enzymes within coacervates and the apparent terpolymer-specific association of TnaA? (Figs. 4c & S14). Do the aggregates contain both enzymes or do they phase separate within the coacervate?

Figure 5b – the inclusion of GFP36+ could be informative here although the aggregation behaviour presumably slows cleavage and diffusion out of the coacervate. While the GFP30- results are very

convincing, the only 50% release of sfGFP raises the question of whether this is an inherent limitation of the strategy or just a result of the conditions of this experiment. Is full release observed after more than one hour or with higher TEV protease concentrations? Is this timescale relevant for the envisioned applications of this system or will further optimisation be required?

Lines 275-278 – This section seems to neglect investigations on enzyme packing in protein-based encapsulation systems, such as virus-like particles or bacterial microcompartments, some relevant literature should be cited in this context.

Minor comments

Line 60 – distinction between 'amount taken up' and 'local concentration' is needed here, as in the current form one seems redundant

Lines 131-135 – this concept appears out of the blue and introduces some confusing details, it may be better placed in the discussion

Line 177 – 'of the enzymes' -> 'of enzyme'

Line 178 – 'slightly functionalized' is vague, would be better to show the degree of labelling

Line 168 – 'the name of the PLP cofactor is misspelled

Line 180 – delete 'was'

Line 218 – could --g> would

Lines 446-448 – the actual degree of labelling should be shown

Line 477 – Reference 1 has an apostrophe after the first author's surname

SI, Figure S3 – the drop in fluorescence above concentrations of 7.5uM Ni²⁺ should be explained

SI, Figure S11 – this figure is redundant and could be removed

Reviewer #2 (Remarks to the Author):

Review of "Programmed spatial organization of biomacromolecules into discrete, coacervate based protocells". Altenburg et al. present a well written paper describing the inclusion of GFP of different charges and anhydase and monooxygenase by NiNTA-His interactions into coacervate based protocells. The authors modify an existing membrane bound coacervate, which they have developed, to chemically incorporate NiNTA groups to a coacervate forming component, amylose, to generate NTA-Am. His-tagged proteins were added to the NTA-Am prior to protocell formation. The authors characterized the amount of his tagged proteins incorporated into the protocell vs non-his tagged proteins; measured the diffusion coefficient of the proteins by FRAP; characterized enzyme activity via a two-step enzyme cascade incorporated into the protocell by the described method and demonstrated the release of protein by TEV cleavage. They authors have presented an new and interesting system. The authors have explicitly described this system as having the ability to incorporate biological functionality with programmed behavior. However, this has not been demonstrated and the promise of the system has not been realized. For this work to be suitable for the readership of nature communications additional experiments and controls to

support some of current observations and conclusions would be required. In addition, true programmability and biological function should also be demonstrated for the impact if the true impact of this system is to be realized.

General comments:

1. How does the addition of the NiNTA to Am and the addition of the different charged proteins affect coacervate formation and their physical properties? These would be important to characterize and include as this may lead to a route to control enzyme activity as described.
2. The authors discuss the ability to have a high level of control over the system. However, this has not really been shown. A very interesting figure is supplementary figure s3, here the authors show that increasing NiNTA modifications increases the amount of GFP into the protocell. Could this approach be used to control the concentration of enzymes anhydrase and monooxygenase to regulate the kinetics? This would support their claims of a "programmed co-localization". Furthermore, with increasing Ni-NTA the plot shows a decrease in the amount of GFP encapsulated within the protocells. This is an interesting observation and some additional words/ experiments to describe this phenomena would help the reader to understand the system further.
3. The authors state that the NiNTA allows the same stoichiometry of the enzymes 1:2 and attributes this to the interactions of His to Ni/NTA. It would be good to include an experiment which shows the stoichiometry of the enzyme within the coacervate based protocell with no NiNTA modifications. Is the stoichiometry maintained here? This would demonstrate that the conservation of enzyme stoichiometry is conserved by the NiNTA-His interactions.
4. The authors say that there is a 150x more enzyme within the protocell with the addition of the Ni/NTA groups. However, other authors from the field have also described the same or more upconcentration within coacervates with no chemical modifications. Is this significant because the sfGFP ordinarily does not go into the coacervates? If this were the case perhaps the authors could make this significance more clear.
5. The authors describe enhanced enzyme activity inside the coacervates however, it would only be clear how enhanced the activity was by comparing the rates of reaction of his-tagged and non-his tagged proteins in buffer and within the coacervate. More experiments along these lines are required to support their assertions here.
6. The authors say in their discussion that "we have designed a new methodology for the introduction of biological function in protocells." It is not clear what the biological function, which has been included here. As it currently stands the manuscript shows they can add in proteins and enzymes but it is not clear where the biological function is. Indeed, this has also been shown previously as highlighted in the introduction. Additional experiments which show the true potential of the system would add further strength to this work making it appealing to the general audience of nature communications.
7. Further to comment 3 and 6, the authors claim they have programmed behavior into the protocell by maintaining the stoichiometry of the enzymes. Could they give a clearer demonstration of programmed behavior?
8. The results with TEV cleavage are very interesting. Is this effect controllable? i.e. by tuning the TEV concentration can the cleavage and kinetics of release be controlled. Does the TEV cleavage begin as soon as the protocell is generated, what triggers the cleavage and how could this be regulated? How do the diffusion coefficients change with and without TEV cleavage, this information could be interesting to characterize the change in coacervate properties due to the inclusion of TEV and enzyme activity.
9. Figure 3c, 4c and 5d, S9, S14 show regions of high fluorescence intensity within the membrane. The authors attribute this to salt-induced protein aggregation. Why would the salt concentration in Figure 3c be different between the different GFP. What are the material properties of the regions of high fluorescence? Would these "aggregates" affect enzyme activity? A more distinct explanation of this observation is required as they seem to be appearing in multiple scenarios and with different proteins.

Minor comments:

10. On lines 104-105, the authors say, "Importantly, protein uptake into the coacervate core is homogeneous, and independent of protocell size." Doesn't the data show that it is dependent on size, there is a linear relationship between the GFP concentration and the size of the protocell?

11. Following from point 8, has figure 2c been normalized for the size of the droplet? As the data in figure 2c seems to show a correlation between the size and concentration.

12. Please can the authors add additional explanation for the data in figure S5 obtained by FACS. It is unclear what the red data points are correlating to compared to the black data points

13. Does the diffusion coefficient, of 30GFP, measured by FRAP change after TEV cleavage? Could this be used to confirm that the GFP is cleaved but stays within the protocell? How does the data reflect that cleavage by TEV takes place here. If the protocell cores are different, it could be that TEV is active in some protocells (containing sfGFP) but not others (30GFP).

14. The authors state that the five-fold increase in the rate for the TnaA:FMO reaction due to an increased local concentration and co-localization. They go on to say that the enzyme activity is independent of enzyme concentration. This seems to be contradictory. An experiment could be undertaken to prove this point, i.e. have a low concentration of enzymes in the coacervate controlled by the NiNTA concentration comparable to the buffer so concentration effects are equal?

Response to reviewers

Original comment

Our response

Original text

Changed text

Reviewer #1 (Remarks to the Author):

The manuscript by Wiggert et al. describes the formation of synthetic protocells that are capable of controlled loading and release of protein cargo. The protocells are based on coacervates that are formed by mixing of positively charged and negatively charged modified amyloses and then stabilized with a synthetic block copolymer coating. The inclusion of an NTA-modified amylose in the coacervate mixture enables the Ni-mediated sequestration of His-tagged proteins regardless of their surface charge, allowing the enhancement of a two-enzyme cascade reaction. Through the use of a cleavable His-tag, cargo proteins can also be released by enzymatic treatment.

Critique

The authors provide a very clear overview of the goals, current limitations and challenges in the area of synthetic protocells, which places this contribution into its broader context quite well. The specific issues with current protocell mimics and the way in which this work addresses them are plainly stated. Overall the manuscript is well-written, logical and easy to follow. The derived conclusions are supported by the data provided and the experimental work is comprehensive and of high quality in both execution and presentation. The general strategy reported seems promising for further development to increase the complexity and functionality of synthetic protocells. For these reasons, this manuscript should be of interest to the broad readership of Nature Communications and be ready for publication after some minor adjustments.

We thank the reviewer for their positive feedback, and for their suggestions below.

Specific comments

Charge ratio – the need for a 2:1 ratio of Q-Am to Cm-Am is unclear in this manuscript and should be explained briefly at the beginning of the results section. It seems that some residual positive charge is needed to drive interfacial assembly of the glutamate-containing terpolymer, but this could be stated explicitly. Confounding this issue are the different degrees of substitution for the Q-Am and Cm-Am, which

are 0.88 and 0.4, respectively. Are the mixtures used for coacervate formation based on molar or charge ratios?

Indeed, there is an excess of positive charge required for stability and terpolymer anchoring. This has been determined in previous work, Mason et al. 2017 JACS. To clarify this for readers interested in this detail, an additional sentence has been added to the methods. Regarding the ratio of amyloses, the main text and methods have been adjusted for clarification.

Methods: *“Consecutively, CM-Am, Q-Am and NTA-Am were added to induce coacervation in a 2:0.8:0.2 mass ratio of Q-Am:CM-Am:NTA-Am. While different ratios of modified amyloses can be used, the mass ratio of 2:1 Q:CM, corresponding to approximately a 3:1 charge ratio due to differing degrees of substitution, has been found to be the most stable.”*

Main text: *“sfGFP-His was mixed during protocell formation with the amylose polymers (2:0.8:0.2 mass ratio of Q:CM:NTA)”*

Figure 2b – the authors state there is a linear correlation between particle volume and fluorescence intensity. While the curve is approximately linear at lower values, it seems to deviate above 50000 a.u. revealing a plateau. For this reason, it would be worth acknowledging this in the main text. More crucially it would then be important to work in the linear range for subsequent experiments, specifically those quantifying different cargo loading and release (Figs. 3b, 4d, 5b).

The size of our protocells varies due the nature of coacervate growth, although the majority of the protocells is between 15-25 μm , larger ones do occur. For the flow cytometer that we have used, with a nozzle size of 70 μm , the particles larger than 30 μm are likely to be influenced by it. Normally for flow cytometry the maximum cell/particle size is between 0.3 to 0.5 times the nozzle size.¹ This can explain the deviation observed above 50000 a.u.. We have acknowledged this by adding the following to the main text: *“At larger size there does appear to be a deviation in the linear behavior, this is likely due to the fact that these larger protocells measured exceed the recommended instrumental limits.”³⁸*

In any case, for the original analyses only protocells in the linear range, between 10-30 μm , were analysed in subsequent experiments (including the cargo loading and release).

Punctate structures are observed for some of the encapsulated proteins (e.g. GFP36+). The authors conclude that this is due to salt-induced aggregation. Are these aggregates present before incubation with coacervates, are they only observed within the protocells, and could they be mediated by free Cm-Am, terpolymer or NTA-Am chains?

Lawrence et al.² already describe that the solubility of +36GFP is heavily dependent on salt concentrations. However the aggregation is reversible, meaning that the protein does not unfold during aggregation. When these proteins are added to the coacervate mixture, their buffer changes from a high salt (600 mM) to PBS (~150

mM), and this change in ionic strength causes the protein to aggregate. To more clearly visualize this, +36GFP was diluted in either PBS or storage buffer. Only in the case of PBS the protein aggregates, in the storage buffer the protein remains in solution.

The figure has also been added to SI for the interested reader (Figure S10).

Although the FRAP experiments are not shown for GFP36+, the diffusion is presumably slowed, this seems to be missing from the discussion in lines 136-149 and should be addressed.

Due to this aggregation of +36GFP, FRAP analysis has not been performed on this protein. This has been clarified in the main text: *“Due the aggregated state of loaded +36GFP, it was not included for these experiments.”*

Quantification of enzyme co-encapsulation. The authors state that the ratio of the two enzymes can be tightly controlled, which is determined by microscopy. However, since both enzymes are labelled with Cy5, it is currently unclear how they could be distinguished when co-encapsulated.

In retrospect, this could have been explained more clearly. Both enzymes were labelled with Sulfo-Cy5 to facilitate their direct comparison. For the analysis different populations of protocells were made where only one of the enzymes was labelled. Different dyes could have been used but this complicates the analysis due to different photophysical properties. To make this clearer, we have added the following to the main manuscript:

“In order to quantify the absolute loading efficiency of each enzyme, two separate populations were prepared: Sulfo-Cy5-NHS labelled TnaA with unlabeled FMO, and Sulfo-Cy5-NHS labeled FMO with unlabeled TnaA.”

Can the authors also rationalise the aggregated structures of both enzymes within coacervates and the apparent terpolymer-specific association of TnaA? (Figs. 4c & S14). Do the aggregates contain both enzymes or do they phase separate within the coacervate?

The labelled enzymes tend to aggregate upon unspecific modification of the lysine residues. This is more pronounced when the enzymes are brought in close proximity, e.g. loaded into the coacervates. By bright-field microscopy the difference can be observed, aggregates are visible inside the coacervates when adding a labelled variant for one of the enzymes.

The figure has also been added to SI for the interested reader (Figure S16).

Figure 5b – The inclusion of GFP36+ could be informative here although the aggregation behaviour presumably slows cleavage and diffusion out of the coacervate.

This was initially not addressed due to the aggregative behavior of the protein due to the lower salt concentration. Nonetheless, we have addressed it and performed TEV cleavage on the +36GFP. After one hour of incubation, a difference can be observed compared to no TEV. Although a clear distinction can be made between protocells, the protease doesn't seem to be able to cleave in every protocell, presumably due to the presence of inaccessible TEV cleavage sites hidden in aggregates.

The figure has also been added to SI for the interested reader (Figure S24).

While the GFP30- results are very convincing, the only 50% release of sfGFP raises the question of whether this is an inherent limitation of the strategy or just a result of the conditions of this experiment. Is full release observed after more than one hour or with higher TEV protease concentrations?

The reviewer raises an interesting question. Therefore, we added an experiment to investigate the release for more than one hour. As can be seen in the figure below, the release approximates >90% after more than two hours.

The figure has also been added to SI for the interested reader (Figure S28).

It this timescale relevant for the envisioned applications of this system or will further optimisation be required?

This is fully dependent on the application; some might require a burst release while in other cases a gradual release over time is desired. This property can be tuned by the TEV concentration. For future work it will be interesting to investigate a triggered activation of this enzyme or an alternative release mechanism.

Lines 275-278 – This section seems to neglect investigations on enzyme packing in protein-based encapsulation systems, such as virus-like particles or bacterial microcompartments, some relevant literature should be cited in this context.

The reviewer makes a valid point here, and we have amended this paragraph in the discussion to include a sentence detailing this phenomenon in protein-based compartments and provided two additional references:

“This substrate bottleneck has also been observed in synthetic enzyme-loaded protein-based compartments, where high local catalyst concentrations do not lead to higher rates of substrate turnover.^{39,50} In future investigations, this limitation could be

overcome by taking inspiration from nature, such as employing active uptake or carrier molecules.”

Minor comments

Line 60 – distinction between ‘amount taken up’ and ‘local concentration’ is needed here, as in the current form one seems redundant

“The modified coacervate droplet was capable of efficient and homogenous sequestration of different recombinant proteins, and allowed for both control over the amount taken up and an increased local concentration.”

Changed to:

“The modified coacervate droplet was capable of efficient and homogenous sequestration of different recombinant proteins, and enabled control over their local concentration.”

Lines 131-135 – this concept appears out of the blue and introduces some confusing details, it may be better placed in the discussion

We agree with the reviewer on this point and have made the following changes to the main text (instead of moving it to the discussion). We have also added several references providing details of biological examples.

“For many proteins function is directly related to diffusivity, for example in signaling cascades and multistep catalytic processes. A crowded environment enables the mimicry of diffusional processes as they occur in an intracellular milieu.”

Changed to:

“Next, we investigated the diffusivity of encapsulated cargoes, to ensure that our programmed uptake strategy does not eliminate cargo mobility. This is an important parameter, because for many proteins function is dependent on their spatiotemporal organization, for example in signalling cascades and multistep catalytic processes.^{40-42.”}

Line 177 – ‘of the enzymes’ -> ‘of enzyme’

“To enable visualization and quantification of the enzymes uptake, both were slightly functionalized with Sulfo-Cy5-NHS.”

Changed to:

“To enable visualization and quantification of enzyme uptake, both were slightly functionalized with Sulfo-Cy5-NHS”

Line 178 – ‘slightly functionalized’ is vague, would be better to show the degree of labelling

“To enable visualization and quantification of enzyme uptake, both were slightly functionalized with Sulfo-Cy5-NHS.”

Changed to:

“To enable visualization and quantification of enzyme uptake, both were slightly functionalized with Sulfo-Cy5-NHS, resulting in a ~0.5 degree of labeling for both.”

Line 168 – ‘the name of the PLP cofactor is misspelled

*“Tryptophan anhydrase (TnaA)⁴⁸ is responsible for the conversion of l-Trp to indole with **peridoxal-5-phosphate** (PLP) as cofactor.”*

Changed to:

*“Tryptophan anhydrase (TnaA)⁴⁸ is responsible for the conversion of l-Trp to indole with **pyridoxal-5-phosphate** (PLP) as cofactor.”*

Line 180 – delete ‘was’

*“The larger size of the enzymes, a 220 kDa tetramer and 111 kDa dimer, for TnaA and FMO respectively, did not hamper efficient sequestration **as was shown** with confocal microscopy (Figure 4c).”*

Changed to:

*“The larger size of the enzymes, a 220 kDa tetramer and 111 kDa dimer, for TnaA and FMO respectively, did not hamper efficient sequestration **as shown** with confocal microscopy (Figure 4c).”*

Line 218 – could --g> would

*Thus, if the His-tag can be cleaved from the sfGFP-His cargo, excretion of the protein from the protocell system **could** be expected (Figure 5a).*

Changed to:

*“Thus, if the His-tag can be cleaved from the sfGFP-His cargo, excretion of the protein from the protocell system **would** be expected (Figure 5a).”*

Lines 446-448 – the actual degree of labelling should be shown

Adjusted this part of the methods section:

The average labelling per protein was measured using the absorption of the dye at 640 nm and at 280 nm for the protein, TnaA-Cy5 degree of labelling: 0.48. FMO-Cy5 degree of labelling: 0.56.

Line 477 – Reference 1 has an apostrophe after the first author’s surname

This is the correct spelling of the author’s surname.

SI, Figure S3 – the drop in fluorescence above concentrations of 7.5 μM Ni^{2+} should be explained.

We agree with the reviewer, an extra explanation was added to the figure caption:

“Decrease in uptake at Ni^{2+} concentrations above 7.5 μM can be explained due to the excess of the ion compared to possible binding sites. The free nickel can still interact with the His-tagged proteins and therefore influence the uptake behavior.”

SI, Figure S11 – this figure is redundant and could be removed

We disagree with the reviewer, the graph shows the fitted exponential decay function over the data points. Additional information on the graph has been added to the caption:

“The FRAP analysis of sfGFP-His (green), -30GFP-His (blue) and -30GFP (open blue circles) with the fitted curves was used to calculate the apparent diffusion constant, main text Figure 3d.”

Reviewer #2 (Remarks to the Author):

Review of “Programmed spatial organization of biomacromolecules into discrete, coacervate based protocells”. Altenburg et al. present a well written paper describing the inclusion of GFP of different charges and anhydase and monooxygenase by NiNTA-His interactions into coacervate based protocells. The authors modify an existing membrane bound coacervate, which they have developed, to chemically incorporate NiNTA groups to a coacervate forming component, amylose, to generate NTA-Am. His-tagged proteins were added to the NTA-Am prior to protocell formation. The authors characterized the amount of his tagged proteins incorporated into the protocell vs non-his tagged proteins; measured the diffusion coefficient of the proteins by FRAP; characterized enzyme activity via a two-step enzyme cascade incorporated into the protocell by the described method and demonstrated the release of protein by TEV cleavage. The authors have presented a new and interesting system. The authors have explicitly described this system as having the ability to incorporate biological functionality with programmed behavior. However, this has not been demonstrated and the promise of the system has not been realized. For this work to be suitable for the readership of nature communications additional experiments and controls to support some of current observations and conclusions would be required. In addition, true programmeability and biological function should also be demonstrated for the impact if the true impact of this system is to be realized.

We thank the reviewer for both their positive and constructive comments. Some excellent points have been made, which have been clarified below.

General comments:

1. How does the addition of the NiNTA to Am and the addition of the different charged proteins affect coacervate formation and their physical properties? These would be important to characterize and include as this may lead to a route to control enzyme activity as described.

The reviewer makes a valid point, it is likely the coacervate core is influenced by the introduction of NiNTA or a protein, since it introduces additional interactions. However, with current experimental techniques it is extremely difficult (if not impossible) to determine parameters such as the concentration of counter-ions, local pH and local stiffness on the micron scale. It is also important to note that the molar fraction of encapsulated species is only a fraction of the overall amylose concentration (proteins – nanomolar; NTA-Am – micromolar; Q/CM-Am – millimolar). As such it is unlikely that any encapsulated species would have a significant influence on the coacervate physical properties.

2. The authors discuss the ability to have a high level of control over the system. However, this has not really been shown. A very interesting figure is supplementary figure s3, here the authors show that increasing NiNTA modifications increases the amount of GFP into the protocell. Could this approach be used to control the concentration of enzymes anhydrase and monooxygenase to regulate the kinetics? This would support their claims of a “programmed co-localization”.

Yes, it would be possible to change the uptake efficiency by varying the amount of NiNTA while keeping the enzyme concentration the same. However, this would result in two populations of active enzyme – those bound to the coacervate core by the NiNTA interaction, and those free in solution. Subsequent analysis of the rates of reaction would not be able to distinguish these two populations without purification and removal of non-encapsulated species, which is why we have elected to control the enzyme concentration in the manner already described.

Furthermore, with increasing Ni-NTA the plot shows a decrease in the amount of GFP encapsulated within the protocells. This is an interesting observation and some additional words/ experiments to describe this phenomena would help the reader to understand the system further.

To clarify, in Figure S3, only the Ni²⁺ concentration is being varied. However we agree with the reviewer that this is interesting behavior, which was also raised by reviewer 1. We have thus added an explanation to the figure caption. It may be helpful to the reviewers to think of the analogous situation with a Ni-NTA column for protein purification. Before any His-tagged protein is loaded, excess unbound Ni²⁺ is removed via extensive washing steps. If this step was not performed, remaining Ni²⁺ could bind His-tagged proteins and prevent their binding to the column.

“Decrease in uptake at Ni²⁺ concentrations above 7.5 μM can be explained due to the excess of the ion compared to possible binding sites, as free Ni²⁺ can still interact with the His-tagged proteins and therefore influence the uptake behavior. This

situation can be compared to immobilized metal affinity chromatography, where extensive washing after metal loading is performed prior to purification.”

3. The authors state that the NiNTA allows the same stoichiometry of the enzymes 1:2 and attributes this to the interactions of His to Ni/NTA. It would be good to include an experiment which shows the stoichiometry of the enzyme within the coacervate based protocell with no NiNTA modifications. Is the stoichiometry maintained here? This would demonstrate that the conservation of enzyme stoichiometry is conserved by the NiNTA-His interactions.

The reviewer has a valid point, therefore additional experiments were performed. Without the Ni present, uptake of FMO is observed, but core-specific sequestration of TnaA is not. Therefore, without the Ni²⁺ present the stoichiometry between the two enzymes is not conserved.

The figure has also been added to SI for the interested reader (Figure S17).

4. The authors say that there is a 150x more enzyme within the protocell with the addition of the Ni/NTA groups. However, other authors from the field have also described the same or more upconcentration within coacervates with no chemical modifications. Is this significant because the sfGFP ordinarily does not go into the coacervates? If this were the case perhaps the authors could make this significance more clear.

Yes, the 150x uptake of sfGFP into these coacervates is remarkable because they are excluded under normal conditions (without His-tag or NiNTA). Nevertheless, we agree with the reviewer, and this could be better described in the main text.

“This concentration effect yields a ~150 fold increase in local concentration, highlighting the unique ability to specifically localize and concentrate functional cargoes, that otherwise cannot be incorporated without extensive modification, via this targeted Ni^{2+} -NTA/His interaction.”

5. The authors describe enhanced enzyme activity inside the coacervates however, it would only be clear how enhanced the activity was by comparing the rates of reaction of his-tagged and non-his tagged proteins in buffer and within the coacervate. More experiments along these lines are required to support their assertions here.

It is not clear what these kinds of experiments would add, however we do agree that the importance of the Ni-His interaction would be highlighted more in the context of enzymatic activity. We decided to carry out some new experiments in which the ratio of TnaA:FMO was varied (with a constant FMO concentration) in both the coacervates and in solution. From these data we were able to draw some interesting conclusions which would not have been possible without the specific uptake mechanism. A new section of text has been added to the main text in the section “Enhanced enzymatic activity in coacervates” with corresponding supplementary data, and the wording of the conclusive statements have been altered accordingly.

“To study the effect of programmed co-localization on this reaction, NADPH consumption and the indoxyl production were followed over time via absorption and fluorescence respectively. Within three hours all the NADPH was consumed by the enzymes sequestered in the coacervates (orange), compared to 15 hours for the same number of enzymes in solution (gray); a five-fold increase in reaction rate was observed (Figure 4f). Moreover, the indoxyl fluorescence intensity did not reach the same level inside the coacervates as in solution, showcasing the faster conversion of intermediate (Figure S13). The higher rates of both reactions are to be attributed to the increased local concentration and co-localization of the enzymes involved due to the Ni^{2+} -NTA/His interaction. The enhancement in enzymatic activity, which is independent of the enzyme concentration, would not be possible without the bio-orthogonal uptake of these enzymes into the coacervate core. These data highlight the unique strength of this platform to study complex enzymatic cascades in confined, discrete, cell-mimetic environments, without significant or detrimental modification of the targeted proteins.”

Changed to:

“To study the effect of programmed co-localization on this reaction, NADPH consumption and the indoxyl production were followed over time via absorption and fluorescence respectively. Within ~3.5 hours all the NADPH was consumed by the enzymes sequestered in the coacervates (orange), compared to >12 hours for the same number of enzymes in solution (gray); a clear increase in reaction rate was observed (Figure 4e). By decreasing the total amount of enzyme for the 0.5:1 ratio by

half, the reaction also takes twice as long, from ~3.5 to ~7 hours. Meanwhile in solution a much lower rate is observed (Figure 4f). Moreover, for both concentrations of enzyme, the indoxyl fluorescence intensity did not reach the same level inside the coacervates as in solution, showcasing the faster conversion of the intermediate (Figure S18). The potential of programmed cargo uptake can be demonstrated by investigating different ratios of loaded enzymes. Over a wide range of TnaA:FMO ratios, the overall rate stays the same with a constant amount of FMO (Figure S19), confirming that FMO is the rate limiting enzyme. In solution however, a clear distinction between the different ratios can be made, where there is a clear dependency on the amount of TnaA present for the reaction rate (gray)(Figure S20). Only when the system is pushed by a large excess of TnaA, (2:1 TnaA:FMO), the rate in solution becomes similar to that in the coacervate system. This type of analysis would not be possible without a robust programmable uptake strategy. Interestingly, in the absence of Ni^{2+} , the time for complete NADPH consumption was similar to Figure 4e (Figure S20). This was unexpected as TnaA has a similar theoretical isoelectric point to GFP (6.19 and 6.04, respectively) and thus should be excluded from the coacervate core. Confocal microscopy indicated non-specific TnaA adsorption on the periphery of the protocells, accomplishing co-localization in the same microenvironment but without the control over the enzyme stoichiometry (Figure S21). This illustrates the need for a generally applicable, bio-orthogonal uptake strategy, which delivers control over the local concentration and co-localization of the enzymes involved. These data represent an important progression towards the study of complex enzymatic cascades in confined, discrete, cell-mimetic environments.”

Figure has been added to the SI for the interested reader (Figure S19)

6. The authors say in their discussion that “we have designed a new methodology for the introduction of biological function in protocells.” It is not clear what the biological function, which has been included here. As it currently stands the manuscript shows they can add in proteins and enzymes but it is not clear where the biological function is. Indeed, this has also been shown previously as highlighted in the introduction. Additional experiments which show the true potential of the system would add further strength to this work making it appealing to the general audience of nature communications.

We respectfully disagree with the reviewer here, as in our view enzymatic activity is biological functionality. The indigo pathway is synthetic in this case because the two enzymes used come from two different organisms, however, in nature similar pathways for indigo production do exist.³ Furthermore it is important to note that the majority of enzymatic cascades shown previously in the literature utilize glucose oxidase and horseradish peroxidase. These enzymes are extremely robust, and as such are extensively used to model two-enzyme cascades in synthetic systems where the functionalization or encapsulation protocol would otherwise destroy activity of a less robust cascade. The aim of this manuscript is to communicate a new, effective, biocompatible, and general methodology for the loading of interesting biomacromolecules and we feel that our results effectively demonstrate these points. For example, the TEV protease used in Figure 5 is a commercially available His-

tagged enzymes – we use it directly from the bottle and apply it successfully in our system without further modification of the enzyme itself or our methodology.

7. Further to comment 3 and 6, the authors claim they have programmed behavior into the protocell by maintaining the stoichiometry of the enzymes. Could they give a clearer demonstration of programmed behavior?

We disagree with the reviewer, we do not claim we have programmed behavior. In this manuscript we claim we have programmed uptake/loading of the cargo. In other words, we control the ratio and amount of cargo that gets taken up into our system. In any case, we have performed additional experiments to show we can program the loading of our protocell platform, e.g. different amounts of TEV (Figure S27) as well as a lower enzyme concentration and different ratio's (Figure 4f and S20).

8. The results with TEV cleavage are very interesting. Is this effect controllable? i.e. by tuning the TEV concentration can the cleavage and kinetics of release be controlled.

Yes, as the new results shown below demonstrate, the kinetics of release can be tuned depending on the amount of TEV protease added initially.

This figure has been added to the SI for the interested reader. (Figure S27)

Does the TEV cleavage begin as soon as the protocell is generated, what triggers the cleavage and how could this be regulated?

In this case there is no control over the protease activity, only over the localization. The protease in this case is inherently active. For future work one could think of light or specific chemical triggers to activate the protease.

How do the diffusion coefficients change with and without TEV cleavage, this information could be interesting to characterize the change in coacervate properties due to the inclusion of TEV and enzyme activity.

This has already been described in Figure 5c. The removal of the His-tag influences the diffusivity in an expected manner.

9. Figure 3c, 4c and 5d, S9, S14 show regions of high fluorescence intensity within the membrane. The authors attribute this to salt-induced protein aggregation. 1. Why would the salt concentration in Figure 3c be different between the different GFP.

The +36GFP is expressed in *E. coli*, then purified and stored in a high salt buffer (> 0.5 M NaCl). Under these high-salt conditions, the +36GFP is stable and does not aggregate. However, as soon as the protein is diluted into a buffer containing a much lower salt concentration, such as the PBS used for our protocell formation experiments (~ 150 mM), aggregation rapidly occurs. To clarify this, we have included additional data where the salt-induced aggregation behavior is depicted (see picture above, and new Figure S10).

What are the material properties of the regions of high fluorescence?

The high fluorescence regions are protein aggregates. In any case, no functional assay have been performed on protein that are in this state.

Would these “aggregates” affect enzyme activity? A more distinct explanation of this observation is required as they seem to be appearing in multiple scenarios and with different proteins.

No activity studies were done with the labelled enzymes, as was already stated in the main text. The labelled enzymes have only been used to visualize the uptake.

Figure S16 (shown above at reviewer #1) was added to show the difference by bright-field microscopy to address the difference between labelled and unlabeled enzyme.

Minor comments:

10. On lines 104-105, the authors say, “Importantly, protein uptake into the coacervate core is homogeneous, and independent of protocell size.” Doesn’t the data show that it is dependent on size, there is a linear relationship between the GFP concentration and the size of the protocell?

We respectfully disagree with the reviewer, there is not a correlation between size and concentration but a correlation between size and total amount of protein. In other words, the concentration remains the same.

11. Following from point 8, has figure 2c been normalized for the size of the droplet? As the data in figure 2c seems to show a correlation between the size and concentration.

No, none of the data has been normalized for the droplet size. As explained in comment #10, there is no correlation between size and concentration. Additionally, only the protocell between 10 – 30 μm are taken into account for our analysis.

12. Please can the authors add additional explanation for the data in figure S5 obtained by FACS. It is unclear what the red data points are correlating to compared to the black data points.

Added explanation on in the figure caption: *“Figure S5. Gate settings to select for single particles based on the forward vs the side scatter, selected population shown in red, i.e. droplet containing a single coacervate protocell. Black data points represent droplets containing multiple protocells, terpolymer aggregates or noise. Left) Forward scatter height vs side scatter area. Right) Forward scatter Height vs forward scatter Area.”*

13. Does the diffusion coefficient, of 30GFP, measured by FRAP change after TEV cleavage?

Yes, the diffusion coefficient of -30GFP changes after TEV cleavage. This is described in the main text and shown in figure 5c.

Could this be used to confirm that the GFP is cleaved but stays within the protocell? How does the data reflect that cleavage by TEV takes place here.

Indeed, we have used this methodology to confirm that -30GFP remains in the protocell meanwhile the sfGFP diffuses out after cleavage.

If the protocell cores are different, it could be that TEV is active in some protocells (containing sfGFP) but not others (30GFP).

In figure 5c we show that the diffusivity of -30GFP changes one order of magnitude upon the addition of TEV. This could only be achieved if the His-tag was removed of the -30GFP, therefore the protease must be active. Moreover, as explained earlier (response to comment #1), the protocell core is not likely to be that different when loaded with another cargo.

14. The authors state that the five-fold increase in the rate for the TnaA:FMO reaction due to an increased local concentration and co-localization. They go on to say that the enzyme activity is independent of enzyme concentration. This seems to be contradictory. An experiment could be undertaken to prove this point, i.e. have a low concentration of enzymes in the coacervate controlled by the NiNTA concentration comparable to the buffer so concentration effects are equal?

The reviewer has a good point here regarding the statement “enzyme activity is independent of enzyme concentration”. We made this statement in reference to the original experimental data wherein the overall enzyme concentration was the same,

however the local concentration within the coacervates differs to that in solution due to the uptake mechanism, therefore causing this remarkable enhancement in activity. In hindsight, this statement was limiting and we therefore did some more experiments which provide a more nuanced picture of the system.

Firstly, the ratio of enzymes was varied in both coacervate and solution, with the main conclusion being that within this range of concentrations, the first enzyme doesn't have an effect on the overall rate. On the other hand, in solution a clear effect was observed, with the rate tending towards that seen in confined microenvironment at high TnaA concentration (Figure S19). This illustrates the significance of the co-localization effect enhancing the overall rate.

Secondly, if the overall concentration of enzymes is halved (Figure 4f), we still observe that the rate is faster in the coacervates, but also that the rate is slower compared to Figure 4e. In this case there is a dependency of the enzyme concentration on overall rate, which highlights the programmable nature of protein uptake in this system.

References:

1. Cossarizza, A. *et al.* Guidelines for the use of flow cytometry and cell sorting in immunological studies. *European Journal of Immunology* **47**, 1584–1797 (2017).
2. Lawrence, M. S., Phillips, K. J. & Liu, D. R. Supercharging Proteins Can Impart Unusual Resilience. *J. Am. Chem. Soc.* **129**, 10110–10112 (2007).
3. Fabara, A. N. & Fraaije, M. W. An overview of microbial indigo-forming enzymes. *Appl Microbiol Biotechnol* **104**, 925–933 (2020).

REVIEWER COMMENTS

Reviewer #1 (Remarks to the Author):

The authors have done well to answer the concerns raised by the reviewers, resulting in a clearer and more convincing functional protocell study.

- Based on the data provided, and mirrored concerns from reviewer 2, the claim of homogenous protein packaging could be toned down. Although the microscopy images show that sfGFP and GFP30- are encapsulated homogeneously, the other three cargo proteins exhibit significant aggregation. Although the aggregation is only directly observed for the Cy5-labelled enzymes, there is no way to tell if the unlabeled enzymes also aggregate to some extent – this cannot really be interpreted from the newly added Figure S16. If the aggregation is indeed caused by dye labelling, one suggestion for future studies would be to avoid labelling the proteins with such a bulky hydrophobic dye, but instead use a small hydrophilic dye such as Atto488, or genetically fuse them to a fluorescent protein. While this aggregation may not be an inherent limitation of the method and will be highly guest dependent, it is still an important point that could be better addressed.
- The authors have amended the discussion to include enzyme encapsulation in protein cages, with the addition of two citations. However, the following investigations, which also concern cascade reactions with two enzymes, would be more relevant to this discussion: Jordan et al. Nat. Chem. 2016 8, 179–185; Frey et al. J. Am. Chem. Soc. 2016, 138, 10072–10075.
- Supplementary Figure S12 – is the R2 value for sfGFP (green line) really correct?
- Supplementary Figure S3 caption – Line 144: Q:Am:Cm should read Q:Cm:NTA (or Q-Am:Cm-Am:NTA-Am)

Writing corrections/suggestions:

Line 47: low complexity domains- this is an unclear term, maybe structural domains?

Line 58: There is a thus -> delete a

Line 63: modifications -> remove plural

Line 68: , and enabled -> , enabling

Line 73: scope -> range

Line 299: parameters that needs to be -> parameters that need to be

Line 300: synthetic reconstituted -> reconstituted synthetic

Line 300: substrates/cofactor -> substrate/cofactor

Reviewer #2 (Remarks to Author)

The revised manuscript, with the additional experiments, is a well written and constructed piece of work which shows the ability to regulate uptake of proteins, demonstrates the activity within the synthetic cells and release of the proteins by interactions of a Ni-NTA loaded core with his-tagged proteins.

Please see below for minor comments (in bold is the comments on the rebuttal).

2.

The authors discuss the ability to have a high level of control over the system. However, this has not really been shown. A very interesting figure is supplementary figure s3, here the authors show that increasing NiNTA modifications increases the amount of GFP into the protocell. Could this approach be used to control the concentration of enzymes anhydrase and monooxygenase to regulate the kinetics? This would support their claims of a “programmed co-localization”.

Yes, it would be possible to change the uptake efficiency by varying the amount of NiNTA while keeping the enzyme concentration the same. However, this would result in two populations of active enzyme – those bound to the coacervate core by the NiNTA interaction, and those free in solution. Subsequent analysis of the rates of reaction would not be able to distinguish these two populations without purification and removal of non-encapsulated species, which is why we have elected to control the enzyme concentration in the manner already described.

In this case, you would not need to keep the enzyme concentration the same. If you vary both the NiNTA and the enzyme could this generate a system with varying reactivity?

2. Furthermore, with increasing Ni-NTA the plot shows a decrease in the amount of GFP encapsulated within the protocells. This is an interesting observation and some additional words/ experiments to describe this phenomena would help the reader to understand the system further.

To clarify, in Figure S3, only the Ni₂₊ concentration is being varied. However we agree with the reviewer that this is interesting behavior, which was also raised by reviewer 1. We have thus added an explanation to the figure caption. It may be helpful to the reviewers to think of the analogous situation with a Ni-NTA column for protein purification. Before any His-tagged protein is loaded, excess unbound Ni₂₊ is removed via extensive washing steps. If this step was not performed, remaining Ni₂₊ could bind His-tagged proteins and prevent their binding to the column.

“Decrease in uptake at Ni₂₊ concentrations above 7.5 μM can be explained due to the excess of the ion compared to possible binding sites, as free Ni₂₊ can still interact with the His-tagged proteins and therefore influence the uptake behavior. This situation can be compared to immobilized metal affinity chromatography, where extensive washing after metal loading is performed prior to purification.”

This is still unclear. Please can some clarity be added.

6. The authors say in their discussion that “we have designed a new methodology for the introduction of biological function in protocells.” It is not clear what the biological function, which has been included here. As it currently stands the manuscript shows

they can add in proteins and enzymes but it is not clear where the biological function is. Indeed, this has also been shown previously as highlighted in the introduction. Additional experiments which show the true potential of the system would add further strength to this work making it appealing to the general audience of nature communications.

We respectfully disagree with the reviewer here, as in our view enzymatic activity is biological functionality. The indigo pathway is synthetic in this case because the two enzymes used come from two different organisms, however, in nature similar pathways for indigo production do exist.³ Furthermore it is important to note that the majority of enzymatic cascades shown previously in the literature utilize glucose oxidase and horseradish peroxidase. These enzymes are extremely robust, and as such are extensively used to model two-enzyme cascades in synthetic systems where the functionalization or encapsulation protocol would otherwise destroy activity of a less robust cascade. The aim of this manuscript is to communicate a new, effective, biocompatible, and general methodology for the loading of interesting biomacromolecules and we feel that our results effectively demonstrate these points. For example, the TEV protease used in Figure 5 is a commercially available Histagged enzymes – we use it directly from the bottle and apply it successfully in our system without further modification of the enzyme itself or our methodology.

I agree with the authors that the systems show very elegantly the ability to undertake two step enzyme reactions and TEV cleavage. However, it is quite a leap to say that this is biological functionality. To avoid any ambiguity between their synthetic reaction cascades and TEV cleavage with biological functionality, either the broad statement should be removed or some sentences included for clarity.

9. Would these “aggregates” affect enzyme activity? A more distinct explanation of this observation is required as they seem to be appearing in multiple scenarios and with different proteins.

No activity studies were done with the labelled enzymes, as was already stated in the main text. The labelled enzymes have only been used to visualize the uptake. Figure S16 (shown above at reviewer #1) was added to show the difference by bright-field microscopy to address the difference between labelled and unlabeled enzyme.

The authors show that that the enzymes are aggregated in buffer and within the coacervates and that the enzymes which are part of their reaction cascade are also aggregated. A control experiment showing how the activity is affected by aggregation is required. It would be interesting to see if protein activity is negatively affected by the aggregation.

Original text

Changed text

Our response

Reviewer 1:

• Based on the data provided, and mirrored concerns from reviewer 2, the claim of homogenous protein packaging could be toned down. Although the microscopy images show that sfGFP and GFP30- are encapsulated homogeneously, the other three cargo proteins exhibit significant aggregation. Although the aggregation is only directly observed for the Cy5-labelled enzymes, there is no way to tell if the unlabeled enzymes also aggregate to some extent – this cannot really be interpreted from the newly added Figure S16. If the aggregation is indeed caused by dye labelling, one suggestion for future studies would be to avoid labelling the proteins with such a bulky hydrophobic dye, but instead use a small hydrophilic dye such as Atto488, or genetically fuse them to a fluorescent protein. While this aggregation may not be an inherent limitation of the method and will be highly guest dependent, it is still an important point that could be better addressed.

We agree with the reviewer and thank him for the useful suggestions in future work. All statements on homogeneity have been toned down or made specific for sfGFP.

“The modified coacervate droplet was capable of efficient and homogenous sequestration of different recombinant proteins, and enabled control over their local concentration.”

Changed to:

“The modified coacervate droplet was capable of efficient sequestration of different recombinant proteins, and enabled control over their local concentration.”

Importantly, protein uptake into the coacervate core is homogeneous, and independent of protocell size.

Changed to:

Importantly, sfGFP uptake into the coacervate core is homogeneous, and independent of protocell size.

Secondly, there is either poor control over the final composition of encapsulated species (statistical), or non-biased uptake (membrane free protocells), which are particularly undesirable when attempting to homogeneously encapsulate a large number of different species.

Changed to:

Secondly, there is either poor control over the final composition of encapsulated species (statistical), or non-biased uptake (membrane free protocells), which are particularly undesirable when attempting to efficiently encapsulate a large number of different species.

The authors have amended the discussion to include enzyme encapsulation in protein cages, with the addition of two citations. However, the following investigations, which also concern cascade reactions with two enzymes, would be more relevant to this discussion:

Jordan et al. Nat. Chem. 2016 8, 179–185; Frey et al. J. Am. Chem. Soc. 2016, 138, 10072–10075.

Agreed, Frey et al. has been added. However, in our view Jordan et al. does not match the discussion point on cascade retardation due to substrate availability. Thus we have decided not to include it.

This substrate bottleneck has also been observed in synthetic enzyme-loaded protein-based compartments, where high local catalyst concentrations do not lead to higher rates of substrate turnover^{39,50}.

Changed to:

This substrate bottleneck has also been observed in synthetic enzyme-loaded protein-based compartments, where high local catalyst concentrations do not lead to higher rates of substrate turnover^{39,50,51}.

- Supplementary Figure S12 – is the R2 value for sfGFP (green line) really correct?

Yes, for clarification the graph has been split in two in two without the break on the X-axis to prevent confusion.

- Supplementary Figure S3 caption – Line 144: Q:Am:Cm should read Q:Cm:NTA (or Q-Am:Cm-Am:NTA-Am)

Agreed, the figure caption has been adjusted.

The dependency of the amount of sfGFP (100 nM) loaded inside coacervates on the concentration of Ni²⁺ with an amylose mass ratio of 2:0.8:0.2 of Q:Am:Cm as determined by confocal microscopy.

Changed to

The dependency of the amount of sfGFP (100 nM) loaded inside coacervates on the concentration of Ni²⁺ with an amylose mass ratio of 2:0.8:0.2 of Q:Cm:NTA as determined by confocal microscopy.

Writing corrections/suggestions:

Line 47: low complexity domains- this is an unclear term, maybe structural domains?

The sentence has been adjusted for clarity.

However, while cargo loading within membrane-free protocells is improved compared to membrane-bound systems, the loading mechanism is typically discriminate: there is only selective uptake of components in the coacervate phase when they are modified with a complementary charge or low complexity domains, such as LAF-1²⁷ or elastin²⁸, which limits the general applicability of these platforms²⁷⁻³⁰.

Changed to:

However, while cargo loading within membrane-free protocells is improved compared to membrane-bound systems, the loading mechanism is typically discriminate: there is only selective uptake of components in the coacervate phase when they are modified with a complementary charge or low complexity, intrinsically unstructured regions, such as LAF-1²⁷ or elastin²⁸, which limits the general applicability of these platforms²⁷⁻³⁰.

Line 58: There is a thus -> delete a

There is a thus a clear need to develop a general, biologically compatible, and modular strategy to control the spatial organization of functional biomolecules into the cytosol-mimetic environment, and enable the expansion of life-like functions in bottom-up synthetic cells.

Changed to:

There is thus a clear need to develop a general, biologically compatible, and modular strategy to control the spatial organization of functional biomolecules into the cytosol-mimetic environment, and enable the expansion of life-like functions in bottom-up synthetic cells

Line 63: modifications -> remove plura

Herein we describe a robust method to recruit a broad range of recombinant proteins into coacervate-based protocells without extensive modifications of the cargo, expanding the toolbox of possible biomimetic functionalities.

Changed to:

Herein we describe a robust method to recruit a broad range of recombinant proteins into coacervate-based protocells without extensive modification of the cargo, expanding the toolbox of possible biomimetic functionalities.

Line 68: , and enabled -> , enabling

The modified coacervate droplet was capable of efficient sequestration of different recombinant proteins, and enabled control over their local concentration.

Changed to:

The modified coacervate droplet was capable of efficient sequestration of different recombinant proteins, enabling control over their local concentration.

Line 73: scope -> range

Our strategy for the spatial organization of proteins into discrete self-assembled systems has enabled a broader scope of biologically relevant functionalities, as well as providing a guiding principle of non-covalent binding between the protocellular scaffold and the functional macromolecular cargo to other cell mimetic platforms.

Changed to:

Our strategy for the spatial organization of proteins into discrete self-assembled systems has enabled a broader range of biologically relevant functionalities, as well as providing a guiding principle of non-covalent binding between the protocellular scaffold and the functional macromolecular cargo to other cell mimetic platforms.

Line 299: parameters that needs to be -> parameters that need to be

Line 300: synthetic reconstituted -> reconstituted synthetic

However, such high levels of local enzyme concentration expose new parameters that needs to be accounted for in synthetic reconstituted cascades.

Changed to:

However, such high levels of local enzyme concentration expose new parameters that need to be accounted for in reconstituted synthetic cascades

Line 300: substrates/cofactor -> substrate/cofactor

There is still a limited understanding of substrates/co-factor (e.g. amino acids, nucleotides, NADPH etc.) localization and availability in this system, which were until now not necessary to consider.

Changed to:

There is still a limited understanding of substrate/co-factor (e.g. amino acids, nucleotides, NADPH etc.) localization and availability in this system, which were until now not necessary to consider.

Reviewer 2:

Please see below for minor comments (in bold is the comments on the rebuttal).

2. The authors discuss the ability to have a high level of control over the system. However, this has not really been shown. A very interesting figure is supplementary figure s3, here the authors show that increasing NiNTA modifications increases the amount of GFP into the protocell. Could this approach be used to control the concentration of enzymes anhydrase and monooxygenase to regulate the kinetics? This would support their claims of a “programmed co-localization”.

Yes, it would be possible to change the uptake efficiency by varying the amount of NiNTA while keeping the enzyme concentration the same. However, this would result in two populations of active enzyme – those bound to the coacervate core by the NiNTA interaction, and those free in solution. Subsequent analysis of the rates of reaction would not be able to distinguish these two populations without purification and removal of non-encapsulated species, which is why we have elected to control the enzyme concentration in the manner already described.

In this case, you would not need to keep the enzyme concentration the same. If you vary both the NiNTA and the enzyme could this generate a system with varying reactivity?

For all experiments performed, the NiNTA concentration is always in large excess compared to the amount of protein (NiNTA: micromolar, Protein: nanomolar range). Varying the concentration of the NiNTA would indeed allow for varying amounts of protein present and thus the activity. However, this gives the system another parameter to tune and we do not see the added value of this approach over only varying the amount of protein.

2. Furthermore, with increasing Ni-NTA the plot shows a decrease in the amount of GFP encapsulated within the protocells. This is an interesting observation and some additional words/ experiments to describe this phenomena would help the reader to understand the system further.

To clarify, in Figure S3, only the Ni²⁺ concentration is being varied. However we agree with the reviewer that this is interesting behavior, which was also raised by reviewer 1. We have thus added an explanation to the figure caption. It may be helpful to the reviewers to think of the analogous situation with a Ni-NTA column for protein purification. Before any His-tagged protein is loaded, excess unbound Ni²⁺ is removed via extensive washing steps. If this step was not performed, remaining Ni²⁺ could bind His-tagged proteins and prevent their binding to the column.

“Decrease in uptake at Ni²⁺ concentrations above 7.5 μM can be explained due to the excess of the ion compared to possible binding sites, as free Ni²⁺ can still interact with the His-tagged proteins and therefore influence the uptake behavior. This situation can be compared to immobilized metal affinity chromatography, where extensive washing after metal loading is performed prior to purification.”

This is still unclear. Please can some clarity be added.

“Decrease in uptake at Ni²⁺ concentrations above 7.5 μM can be explained due to the excess of the ion compared to possible binding sites, as free Ni²⁺ can still interact with the His-tagged proteins and therefore influence the uptake behavior. This situation can be compared to immobilized metal affinity chromatography, where extensive washing after metal loading is performed prior to purification.”

Changed to:

From 1 to 7.5μM of Ni²⁺ there is more NTA compared to Ni²⁺, so all Ni²⁺ will be taken up into the coacervate and bind to the NTA. From 7.5μM to 50 μM Ni²⁺, an excess of Ni²⁺ is present compared to NTA. Therefore, there will be increasing amounts of Ni²⁺ in solution which can still interact with the His-tagged proteins. This decreases the driving force for the uptake of the His-tagged protein and thus yields a lower amount localized within the coacervate.

6. The authors say in their discussion that “we have designed a new methodology for the introduction of biological function in protocells.” It is not clear what the biological function, which has been included here. As it currently stands the manuscript shows they can add in proteins and enzymes but it is not clear where the biological function is. Indeed, this has also been shown previously as highlighted in the introduction. Additional experiments which show the true potential of the system would add further strength to this work making it appealing to the general audience of nature communications.

We respectfully disagree with the reviewer here, as in our view enzymatic activity is

biological functionality. The indigo pathway is synthetic in this case because the two enzymes used come from two different organisms, however, in nature similar pathways for indigo production do exist.³ Furthermore it is important to note that the majority of enzymatic cascades shown previously in the literature utilize glucose oxidase and horseradish peroxidase. These enzymes are extremely robust, and as such are extensively used to model two-enzyme cascades in synthetic systems where the functionalization or encapsulation protocol would otherwise destroy activity of a less robust cascade. The aim of this manuscript is to communicate a new, effective, biocompatible, and general methodology for the loading of interesting biomacromolecules and we feel that our results effectively demonstrate these points. For example, the TEV protease used in Figure 5 is a commercially available His-tagged enzyme – we use it directly from the bottle and apply it successfully in our system without further modification of the enzyme itself or our methodology.

I agree with the authors that the systems show very elegantly the ability to undertake two step enzyme reactions and TEV cleavage. However, it is quite a leap to say that this is biological functionality. To avoid any ambiguity between their synthetic reaction cascades and TEV cleavage with biological functionality, either the broad statement should be removed or some sentences included for clarity.

The reviewer has a fair point, this statement has been adjusted in the main text.

In this work, we have designed a new methodology for the introduction of biological function in protocells.

Changed to:

In this work, we have designed a new methodology for the introduction of biologically inspired function in protocells.

9. Would these “aggregates” affect enzyme activity? A more distinct explanation of this observation is required as they seem to be appearing in multiple scenarios and with different proteins.

No activity studies were done with the labelled enzymes, as was already stated in the main text. The labelled enzymes have only been used to visualize the uptake. Figure S16 (shown above at reviewer #1) was added to show the difference by bright-field microscopy to address the difference between labelled and unlabeled enzyme.

The authors show that that the enzymes are aggregated in buffer and within the coacervates and that the enzymes which are part of their reaction cascade are also aggregated. A control experiment showing how the activity is affected by aggregation is required. It would be interesting to see if protein activity is negatively affected by the aggregation.

A control experiment has been performed in which 125 nM of TnaA-Cy5 and 250 nM of FMO-Cy5 have been used, the conditions are the same as in figure 4f. There is a clear difference in activity observed where the labeled enzymes perform worse compared to the unlabeled version, ~10 hours to ~7 respectively. Therefore the aggregated state of the enzymes induced by functionalization with the Cy5-dye affects the protein's activity negatively.

This graph has been added to the SI (Figure S17) for the interested reader.

REVIEWERS' COMMENTS

Reviewer #1 (Remarks to the Author):

The authors have satisfactorily addressed my concerns. Publication can therefore be recommended. This paper will be a fine contribution to Nature Communications

Reviewer #2 (Remarks to the Author):

The authors have addressed all of the comments